# RECAST: Reparameterized, Compact weight Adaptation for Sequential Tasks

**Nazia Tasnim**
Boston University
nimzia@bu.edu

**Bryan A. Plummer**
Boston University
bplum@bu.edu

## Abstract

Incremental learning aims to adapt to new sets of categories over time with minimal computational overhead. Prior work often trains efficient task-specific adapters that modify frozen layer weights or features to capture relevant information without affecting predictions on any previously learned categories. While these adapters are generally more efficient than finetuning the entire network, they still can require tens or hundreds of thousands of task-specific trainable parameters, even for relatively small networks. This is can be problematic for resource-constrained environments with high communication costs, such as edge devices or mobile phones. Thus, we propose **Re**parameterized, **C**ompact weight **A**daptation for **S**equential **T**asks (**RECAST**), a novel method that dramatically reduces the number of task-specific trainable parameters to fewer than 50 – several orders of magnitude less than competing methods like LoRA. RECAST accomplishes this efficiency by learning to decompose layer weights into a soft parameter-sharing framework consisting of a set of shared weight templates and very few module-specific scaling factor coefficients. This soft parameter-sharing framework allows for effective task-wise reparameterization by tuning only these coefficients while keeping the templates frozen. A key innovation of RECAST is the novel weight reconstruction pipeline called Neural Mimicry, which eliminates the need for training in our framework from scratch. Extensive experiments across six diverse datasets demonstrate RECAST outperforms the state-of-the-art by up to $\sim 1.5\%$ and improves baselines $> 3\%$ across various scales, architectures, and parameter spaces. Moreover, we show that RECAST's architecture-agnostic nature allows for seamless integration with existing methods, further boosting performance[1].

## 1 Introduction

*Incremental Learning (IL)* aims to learn from a continuous stream of data or tasks over time. Generally speaking, the goal of IL methods is to avoid catastrophic forgetting (McCloskey & Cohen, 1989; French, 1999) while minimizing computational overhead. As summarized in Figure 1(a), prior work in IL can be separated into three themes. First, *Rehearsal* methods that learn what samples to retain to use on subsequent iterations to ensure their task-specific knowledge is retained (Caccia et al., 2020; Rebuffi et al., 2017; Chaudhry et al., 2019; 2021; Shin et al., 2017), but they can require large amounts of storage as the number of tasks grows. Second, *Regularization* methods that require no extra storage (Kirkpatrick et al., 2017; Kim et al., 2021; Aljundi et al., 2018; Nguyen et al., 2018; Zhang et al., 2020), but can result in network instability due, in part, to the plasticity-stability dilemma (Grossberg, 1982). Finally, third, *Reconfiguration/Adaptation* methods (Yoon et al., 2018; Rusu et al., 2006; Hung et al., 2019; Mallya et al., 2018; Singh et al., 2020; Verma et al., 2021; Jin & Kim, 2022; Ge et al., 2023; Douillard et al., 2022; Wang et al., 2022a) that learn task-specific parameters, but can result in large numbers of new parameters per-task that may not scale well. Even methods designed to be efficient, like those based on LoRA (Hu et al., 2022), still require tens or hundreds of thousands of task-specific trainable parameters for each task (*e.g.*, (Wang et al., 2024; Zhang et al., 2024; Erkoç et al., 2023)) making them ill-suited for some applications in resource-constrained environments like mobile phones or edge devices.

---

[1]Code: Repository

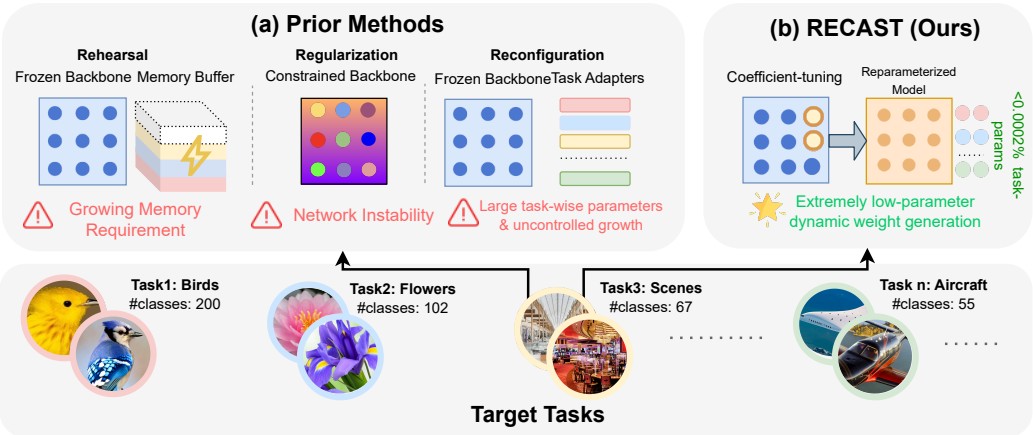

Figure 1: (a) Existing IL methods, *i.e.* Rehearsal, Regularization, Reconfiguration - exhibit various limitations in terms of model complexity, memory requirements, and training overheads. In comparison our proposed method, (b) RECAST can be uses as a frozen backbone, allowing efficient reparameterization of any target module with a negligible number of parameter updates (order of $10^{-6}$) and can accommodate any number of disjoint tasks.

To address this, we propose **Re**parameterized, **C**ompact weight **A**daptation for **S**equential **T**asks (**RECAST**), which provides an extremely parameter-efficient approach for IL via reparameterization of a model's pretrained weights. As illustrated in Figure 1(b), RECAST achieves its compact task-specific representation by finetuning only the coefficients for a set of shared weight templates, enabling us to reconfigure an entire network's weights using fewer than 50 trainable parameters. RECAST framework shares some principles with prior Template Mixing methods (*e.g.*, (Plummer et al., 2022; Savarese & Maire, 2019)). However, these methods did not explore incremental learning or reparameterization, focusing instead on hierarchy discovery and parameter allocation. They also required training from scratch, preventing the use of existing pre-trained models.

In contrast, our proposed **Neural Mimicry** approach allows RECAST to utilize existing large pretrained weights without costly retraining. This weight reconstruction approach learns to decompose pretrained layers into our framework, effectively emulating the original network's weights. This enables us to leverage any pretrained model with minimal processing (often just a few minutes). Thus, REACAST addresses a gap in reparameterization schemes and IL research where reconstruction methods remain underexplored where recent work has focused on generating weights for LoRA, typically through diffusion (Wang et al., 2024; Zhang et al., 2024; Erkoç et al., 2023) or used pretrained weights to initialize their LoRA components (Liu et al., 2024a; Si et al., 2024). Some prior work also learned additional predictor networks from scratch to generate weights for a subset of modules (Salimans & Kingma, 2016). In contrast to these methods, our work directly changes the backbone weights rather than learning an adapter like LoRA-based methods, while also being several orders of magnitude more parameter efficient. Figure 2 provides an overview of our approach.

We evaluate RECAST's effectiveness on diverse datasets in a Task-incremental IL setting. We compare against 16 baselines comprising state-of-the-art IL methods on both CNN and Transformer architectures, where RECAST reports $> 3\%$ gain over prior work. When combined with adapter methods with higher parameter budgets, RECAST further increases performance by $\sim 1.5\%$.

Our key contributions can thus be summarized as follows:

- We propose *RECAST*, A novel framework that dynamically generates module weights using shared basis matrices & module-specific coefficients. This weight-decomposed architecture enables task-specific adaptations optimizing only coefficients, leveraging cross-layer knowledge.
- We introduce *Neural Mimicry*, a novel weight-reconstruction pipeline, that reproduces network weights without resource-intensive pretraining. This scale and architecture-agnostic approach enables easy upgrades to existing architectures. We demonstrate that it builds stronger, more expressive backbones for transfer learning.

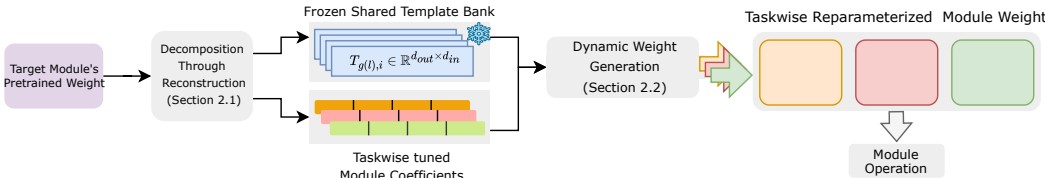

Figure 2: RECAST decomposes module weights into templates and coefficients, which are learned through reconstruction from pretrained weights (Section 3.2). These components are linearly combined to dynamically generate weights for a target layer (Section 3.1). During TIL training, the templates are kept frozen, and only task-specific coefficients are learned to create new layer weights.

- RECAST's fine-tuned resource-performance control, suits efficient continual learning with bounded resource constraints. Its compact task-specific parameters enable easier distribution across large-scale systems and edge devices.

## 2   RELATED WORK

Our work builds upon and extends several key areas in neural network adaptation, in particular the area of weight decomposition. These methods have previously shown that such modular networks are well suited for task adaptation (Mendez & Eaton, 2021; Ostapenko et al., 2021). Older works like Weight Normalization (Salimans & Kingma, 2016) decouples the length of target weight vectors from their direction to reparameterize the network weights. Decomposition has also been used to make DNNs more interpretable (Zhou et al., 2018) and to discover hierarchical structures (Patil et al., 2023). Savarese & Maire (2019) and Plummer et al. (2022) demonstrate the effectiveness of weight sharing through decomposition, similar to our approach. However, these methods either require training models from scratch in their decomposed form or learning additional task-driven priors to accommodate new tasks (Veniat et al., 2020). In terms of parameter-efficient tuning, LoRA is now popularly used to adapt to new tasks through low-rank updates (Chen et al., 2022; Liang & Li, 2024; Zhu et al., 2024; Nikdan et al., 2024), but adds thousands of tunable parameters. In contrast, our approach learns to decompose the weights of a pretrained network to enable quick adaptation to new tasks with $< 0.0002\%$ task parameters.

Our work is also similar to works that generate neural network weights. Earlier works like, HyperNetworks (Ha et al., 2017) uses a small subnetwork to predict the weights of a larger network. Unlike our reconstruction scheme, this requires end-to-end training on input vectors. NeRN (Ashkenazi et al., 2023) builds upon this idea of subnetworks by using positional embeddings of weight coordinates; however, it does not leverage pre-trained knowledge. A recent work, DoRA (Liu et al., 2024b) shares some similarities with our approach by decomposing pretrained weights into magnitude vectors and LoRA matrices. However, the weight is utilized only during initialization and requires significant parameter tuning. Another contemporary work, Wang et al. (2024) attempts novel weight generation through autoencoders and latent diffusion, which can be resource-heavy to train. In comparison, RECAST offers a direct, efficient approach to weight generation and reparameterization. We also show that RECAST can be combined with existing methods to boost performance.

## 3   REPARAMETERIZED, COMPACT WEIGHT ADAPTATION FOR SEQUENTIAL TASKS (RECAST)

In Task Incremental Learning (TIL) a model sequentially learns D tasks $\{d_1, d_2, d_3, \ldots d_D\}$, where each task $d_i = (X_i, Y_i)$ introduces new categories, domains, or output spaces. The objective is to maximize classification performance across all tasks while minimizing resource usage, *i.e.*, $\max \sum_{i=1}^{D} P(d_i)$ s.t. $R \leq R_{max}$, where $P(d_i)$ is the performance on task $d_i$, $R$ is the total resource usage, and $R_{max}$ is the resource constraint. This resource-awareness is crucial in practice, as it enables deployment on edge devices with limited memory, impacts cloud infrastructure costs and real-time processing, and ensures accessibility in resource-limited environments. Thus, TIL poses a complex optimization problem: how can we design a system that adapts to new tasks without significantly increasing its computational footprint or compromising performance on existing tasks?

---

**Algorithm 1** RECAST Framework

---

**Require:**
 $L$: Total number of layers in network
 $G$: Number of template bank groups
 $n$: Number of templates per bank
 $K$: Number of coefficient sets
 $d$: Dimension of weight matrices
 $M_l$: Number of target modules in layer $l$
1:  **procedure** INITIALIZE($L, G, n, K, d, M_l$)
2:   **for** $g = 1$ to $G$ **do**
3:    $\tau_g = T_{g,i} \in \mathbb{R}^{d \times d} | i = 1, \ldots, n$        $\triangleright$ Template bank for group $g$
4:   **end for**
5:   **for** $l = 1$ to $L$ **do**
6:    **for** $m = 1$ to $M_l$ **do**
7:     $C_{l,m} = C_{l,m,i}^k \in \mathbb{R} | k = 1, \ldots, K; i = 1, \ldots, n$   $\triangleright$ Module coefficients
8:     OrthogonalInitialization($C_{l,m}$)
9:    **end for**
10:   **end for**
11: **end procedure**
12: **procedure** FORWARDPASS($x, l, m$)
13:  $g = \lceil l/(L/G) \rceil$                $\triangleright$ Get group index
14:  $W_{l,m} = \frac{1}{K} \sum_{k=1}^{K} \sum_{i=1}^{n} C_{l,m,i}^k \cdot T_{g,i}$      $\triangleright$ Generate weights
15:  $y = \text{ModuleOperation}(x, W_{l,m})$      $\triangleright$ Apply layer operation
16:  **return** $y$
17: **end procedure**

---

We address this multifaceted challenge with our novel framework **Re**parameterized, **C**ompact weight **A**daptation for **S**equential **T**asks (**RECAST**), that strategically balances model plasticity and stability while adhering to resource constraints. As illustrated in Figure 2, RECAST introduces a flexible way to adapt the layers of neural networks through the dynamic combination of sets of shared weight blueprints (*i.e.*, **template banks**) and module-wise calibration factors or **coefficients**. In Section 3.1 we present our Weight Decomposition scheme. Section 3.2 introduces *Neural Mimicry* for direct reconstruction of pretrained weights into reusable templates and coefficients.

## 3.1 WEIGHT DECOMPOSITION

At its core RECAST replaces pretrained layer weights with modules that can dynamically generate weights. A target module has access to a bank of templates which is shared with some other layers in the network (discussed in Section 4.2). Each module has its own set of coefficients which are used to generate the weights. Formally, let $L$ be the total number of layers in the network organized into $G$ groups, where each group shares a template bank. For each group $g \in 1, \ldots, G$, we define a template bank $\tau_g = T_{g,1}, T_{g,2}, \ldots, T_{g,n}$, where $n$ is the number of templates per bank. Each template $T_{g,i} \in \mathbb{R}^{d_{out} \times d_{in} \times \cdots}$ has the same dimensions as the layer being parameterized. For each target module $m \in 1, \ldots, M_l$ in layer $l \in 1, \ldots, L$, where $M_l$ is the number of target modules in layer $l$, we generate $K$ sets of coefficient vectors. The $k$-th set of coefficients for module $m$ in layer $l$ is defined as $C_{l,m}^k = C_{l,m,1}^k, C_{l,m,2}^k, \ldots, C_{l,m,n}^k$, where $C_{l,m,i}^k \in \mathbb{R}$ is the learned coefficient for template $T_{g,i}$ in group $g = \lceil l/(L/G) \rceil$. The weight matrix for module $m$ in layer $l$ is computed via:

$$W_{l,m}^k = \sum_{i=1}^{n} T_{g,i} \cdot C_{l,m,i}^k \tag{1}$$

where $\cdot$ represents element-wise multiplication. The final weight matrix for module $m$ in layer $l$ is generated by averaging these $K$ parameter matrices.

$$W_{l,m} = \frac{1}{K} \sum_{k=1}^{K} W_{l,m}^k \tag{2}$$

---

**Algorithm 2** Neural Mimicry

---

**Require:**
    $M$: Original pretrained model
    $M^*$: Reconstructed Model
    $L$: Number of layers
    $M_l$: Number of modules in layer $l$
    $T_{g(l)}$: Template bank for group $g$ containing layer $l$
    $C_{l,m}$: Coefficients for module $m$ in layer $l$

1:  **procedure** NEURALMIMICRY($M, M^*,$ max_epochs)
2:     **for** epoch $= 1$ **to** max_epochs **do**
3:        total_loss $\leftarrow 0$
4:        **for** $l = 1$ **to** $L$ **do**                                             ▷ Layer iteration
5:           **for** $m = 1$ **to** $M_l$ **do**                                    ▷ Module iteration
6:              $W_{l,m}^* \leftarrow$ GenerateWeights($T_{g(l)}, C_{l,m}$)      ▷ Weight reconstruction
7:              $L_{l,m} \leftarrow$ LossFunction($W_{l,m}^*, W_{l,m}$)
8:              total_loss $\leftarrow$ total_loss $+ L_{l,m}$
9:              $T_{g(l)}, C_{l,m} \leftarrow$ Update($T_{g(l)}, C_{l,m}, L_{l,m}$)
10:          **end for**
11:        **end for**
12:     **end for**
13:     **return** $M^*$                                              ▷ Return reconstructed model
14: **end procedure**

---

Algorithm 1 describes how template banks for each group are created and as well as the orthogonal initialization of coefficients for each module $m$ in layer $l$. The forward pass computes the final weight matrix using these components before applying the module operation. The target module then generates the appropriate output (*e.g.*, convolutional feature map for CNN, attention feature vectors for attention module). See Appendix A.4.1 for additional architecture-specific details.

## 3.2 NEURAL MIMICRY

We introduce "Neural Mimicry" as a foundational technique for efficiently emulating complex pretrained neural networks. This end-to-end pipeline enables the recreation of a target model within our framework, bypassing resource-intensive pretraining. The input to this pipeline is a pretrained target model $M$, whose module weights we aim to emulate. The output is a set of template banks and coefficients that, when combined, approximate the target module weights of $M$. The core of Neural Mimicry lies in its iterative refinement process, which progressively shapes the custom model $M^*$ to closely mirror the behavior of the target model $M$. In our experiments the pretrained weights of M are obtained from official releases (see Appendix A.3). To generate weights for each module in each layer, we employ Equations 1 and 2 as described in Section 3.1, and then aim to reduce the difference between the original pretrained weight ($W_{l,m}$) and RECAST-generated weight ($W_{l,m}^*$). The main update step uses gradient descent (Algorithm 2 line 9) on the coefficients and templates ($\eta$ is the learning rate): $C_{l,m,i}^k \leftarrow C_{l,m,i}^k - \eta \frac{\partial \mathcal{L}}{\partial C_{l,m,i}^k}; T_{g(l),i} \leftarrow T_{g(l),i} - \eta \frac{\partial \mathcal{L}}{\partial T_{g(l),i}}$. The process can be formulated as the following optimization problem:

$$\min_{C,T} \mathbb{E}_{\epsilon \sim \mathcal{N}(0,\sigma^2)} \left[ \sum_{l=1}^{L} \sum_{m=1}^{M_l} \mathcal{L}(W_{l,m}^*(\epsilon), W_{l,m}) \right] \tag{3}$$

where $\epsilon = (\epsilon_1, \ldots, \epsilon_K)$ is a set of K-noise vectors sampled from a Gaussian distribution $\mathcal{N}(0, \sigma^2)$, where $\epsilon_i \in \mathbb{R}^{n \times 1 \times \cdots}$. The noise vectors are added with the coefficients only during the reconstruction training process. Here, $L$ denotes total layers, $M_l$ target modules in layer $l$, $n$ templates per bank, and $\mathcal{L}$ the weight discrepancy loss. As discussed in Section 4.2, $\mathcal{L}$ should be chosen carefully, and we found Smooth L1 loss to be most effective for generating diversified template. It offers smooth gradients for small differences and stability for larger errors, crucial for stable reconstruction (Figures 5 & 4). This outperforms L2/MSE (sensitive to outliers) and KL divergence (computationally expensive) (See Figures 5 & 6). After reconstruction, we calculate the layerwise

feature similarity between $M$ and $M^*$ using the cosine similarity metric to evaluate the quality (see Table 6). Overall, this process requires negligible overhead ($< 5$ minutes, see Table 5).

## 3.3 PARAMETRIC SCALING STRATEGY

RECAST's design is governed by three parameters: the number of groups ($G$), templates per group ($n$), and coefficient sets ($K$), balancing model expressiveness and efficiency. Increasing $G$ improves specialization, while increasing $n$ increases expressiveness within groups. The parameter-sharing scheme allows $K$ to expand weight configurations without significantly increasing parameters. This design achieves substantial memory savings, $\mathbb{S} = L \times M_l \times d^2 - (G \times n \times d^2 \times M_l + L \times M_l \times n \times K)$. Here, $M_l$ refers to the number of target modules per layer, and we can approximate the savings as: $\mathbb{S} \approx L \times M_l \times d^2 - G \times n \times d^2 \times M_l = d^2 M_l (L - G \times n)$. The savings stem primarily from replacing $L \times M_l$ weight matrices with $G \times n \times M_l$ templates. In our experiments, $G$ and $n$ are calibrated to match the baseline model's parameter count but can be adjusted for desired compression. Additionally, multiple coefficient sets enable a combinatorially large number of weight matrices ($n^K$) per module, covering extensive parameter spaces even with small $n$. The total learnable coefficients are $\sum_{l=1}^{L} M_l \times n \times K$, providing flexibility in controlling tunable parameters.

## 4 EXPERIMENTS AND ANALYSIS

**Baselines.** To evaluate our proposed architecture, we have compared the performance against 16 baselines in total, representing the aforementioned categories of IL strategies. From *Regularization* approaches, we include **EWC** (Kirkpatrick et al., 2017), which penalizes changes to critical network parameters, and **LwF** (Li & Hoiem, 2016), which uses previous model outputs as soft labels. For *Replay Methods*, we implement **GDumb** (Prabhu et al., 2020) with a 4400-sample memory buffer, and **ECR-ACE** (Caccia et al., 2022) and **DER++** (Buzzega et al., 2020), both using 120 samples per class. *Reconfiguration* methods are our primary focus as they are most similar to our approach, particularly adapter-based schemes that learn extra task-specific components. Among them, **PiggyBack** (Mallya et al., 2018) uses binary masking schemes within a fixed architecture, while **HAT** Serra et al. (2018) learns a soft attention mask. **CLR** Ge et al. (2023) is a purely adapter-based method, that introduces depthwise separable Ghost modules (Han et al., 2020) after each convolutional layer. We also consider several LoRA-based approaches that apply LoRA to various ViT components (Attention (Zhu et al., 2024) or Fully-connected layers (Chen et al., 2022)). In addition, we compare with **InfLoRA** (Liang & Li, 2024), which has regularization and reconfiguration components, **RoSA** (Nikdan et al., 2024), which combines LoRA with sparse adaptation, **DoRA** (Liu et al., 2024b) that decomposes pretrained weight into magnitude and direction matrices, and a prompt-based adapter technique: **L2P** (Wang et al., 2022b). Finally, we include finetuned classifier layers as naive baselines.

**Datasets.** Following standard experiment setups in similar works by Aljundi et al. (2019); Ge et al. (2023) we employ **six** diverse benchmarking datasets covering fine-grained to coarse image classification tasks across various domain including flowers (Nilsback & Zisserman, 2008), scenes (Quattoni & Torralba, 2009), birds (Wah et al., 2011), animals (Krizhevsky & Hinton, 2009), vehicles (Maji et al., 2013), and other man-made objects (Krizhevsky & Hinton, 2009). These datasets capture the diversity and inter-task variance essential for evaluating IL tasks. Images are standardized to $224 \times 224$ pixels, applying basic transformations and dataset-specific normalization. Additional details are in Appendix A.1 Table 3 .

**Metrics.** We report Top-1 accuracy averaged over our six datasets. For RECAST, we report the mean and standard deviation averaged over 3 runs. See Appendix Table 4 for per-task performance.

**Implementation Details.** We use both ResNet (He et al., 2016) and Vision Transformer (ViT) (Dosovitskiy et al., 2020) backbones to demonstrate the versatility of our framework as they represent two different branches of image classification architectures (CNN and Transformer). Specifically, we use ResNet-34 and ViT-Small variants for all our methods, both of which have approximately 21 $mil.$ parameters. However, average best performance across the six datasets for larger models is also presented in Figure 3. For fair evaluation, we compare the ResNet dependent baselines, against our RECAST-Resnet models and compare ViT-based baselines against RECAST-ViT model. We trained GDUMB (Prabhu et al., 2020), EWC (Lee et al., 2019), LWF (Li

Table 1: TIL accuracy averaged over six datasets using a ResNet-34 comparing non-adapter (top) and adapter methods (bottom). RECAST surpasses non-adapter methods and enhances adapter-based approaches when combined. Parameters: $M$: model size, $R$: buffer, P: total parameters, $U$: unit attention mask, $G$: ghost modules Task Parameters and Storage metrics are reported Per Task

| Method | Task Params | Storage | Acc@top1(%) |
|---|---|---|---|
| ResNet-34 (He et al., 2016) | 0.0 | $M$ | 65.2 |
| EWC (Lee et al., 2019) | $P + FIM$ | $M + FIM$ | 22.1 |
| LWF (Li & Hoiem, 2016) | $P$ | $M$ | 19.3 |
| GDUMB (Prabhu et al., 2020) | P | $M + R$ | 34.8 |
| ER-ACE (Caccia et al., 2022) | $P$ | $M + R$ | 36.1 |
| DER++ (Buzzega et al., 2020) | $P$ | $M + R$ | 16.2 |
| RECAST (ours) | $3 \times 10^{-6}P$ | $M$ | 66.3±0.05 |
| HAT (Serra et al., 2018) | $P$ | $U$ | 40.0 |
| Piggyback (Mallya et al., 2018) | $P$ | $M$ | 82.9 |
| CLR (Ge et al., 2023) | $4 \times 10^{-3}P$ | $M + G$ | 79.9 |
| RECAST (ours) + Piggyback | $P$ | $M$ | **83.9**±0.1 |
| RECAST (ours) + CLR | $4 \times 10^{-3}P$ | $M + G$ | 80.9±0.3 |

& Hoiem, 2016), and L2P (Wang et al., 2022b) using the *avalanche* library (Carta et al., 2023). Official PyTorch implementations of other methods were modified for TIL settings. We evaluated two RoSA (Nikdan et al., 2024) variants: sparse MLP adaptation and full RoSA (LoRA-based attention). A sparsity of $0.001\%$ ensured active parameters remained under 150, matching RECAST's efficiency. LoRA methods used rank=1 and task-specific configurations: DoRA (Liu et al., 2024b) (Q,K,V matrices), MeLo (Q,V matrices), and Adaptformer (MLP layers). RECAST, MeLo (Zhu et al., 2024), and AdaptFormer (Chen et al., 2022), RoSA (Nikdan et al., 2024), DoRA (Liu et al., 2024b) used *AdamW* (Loshchilov & Hutter, 2017) with $2e-3$–$5e-3$ learning rates, $1e-6$ weight decay, and stepwise LR scheduling (decay by 0.1 every 33 epochs) for 100 epochs. Default hyperparameters were used for *avalanche* models and methods like HAT (Serra et al., 2018), Piggyback (Mallya et al., 2018), and CLR (Ge et al., 2023), trained for 100 epochs. InfLoRA (Liang & Li, 2024) used class increments of 90 and 10 epochs per task. For RECAST-ViT, we used a group size of 6, 2 templates per bank, and 2 coefficient sets. Integrated models used RECAST as the backbone, with adapter layers modifying features while RECAST generated core layer weights and coefficients adapted per task (See Appendix A.4.2). All experiments were run on a single RTX8000 GPU.

## 4.1 RESULTS

**Table** 1 showcases the performance of different methods using a ResNet-34 backbone. Traditional IL methods such as EWC (Lee et al., 2019) and LWF (Li & Hoiem, 2016) show significant catastrophic forgetting, with performance drops of approximately $66\%$ and $70\%$ respectively, compared to the baseline ResNet-34. GDUMB (Prabhu et al., 2020), ER-ACE (Caccia et al., 2022), despite utilizing a large memory buffer, only perform slightly better than these traditional methods but still lag significantly behind simple finetuning. HAT (Serra et al., 2018), which learns unit-wise masks over the entire network, achieves further improvement over GDUMB but still lags $39\%$ behind the baseline. This underperformance is likely because HAT was designed to allocate fixed network capacity to each task, struggling with our diverse task suite. In contrast, RECAST improves upon the baseline ResNet-34 by $\sim 2\%$, while fine-tuning only $0.0003\%$ of the original parameters. This represents a significant efficiency advantage while still delivering performance gains. Among adapter methods, Piggyback (Mallya et al., 2018) and CLR (Ge et al., 2023) show strong performance, exceeding the baseline by $27\%$ and $23\%$ respectively. Combined with these methods, RECAST further enhances their effectiveness. RECAST + Piggyback achieves $\sim 1.5\%$ improvement over Piggyback alone. Similarly, RECAST + CLR also shows a $\sim 1.5\%$ gain over standalone CLR. These consistent improvements across different adapter-based approaches demonstrate the versatility and effectiveness of RECAST as a complementary technique.

**Table** 2 summarizes the results for methods using the ViT-Small (Dosovitskiy et al., 2020) backbone, further emphasizing RECAST's strong performance. RECAST improves the baseline ViT-Small by

Table 2: TIL accuracy averaged over six datasets using a ViT-Small comparing non-adapter (top) and adapter methods (bottom). RECAST improves baseline accuracy by 3 points and enhances existing methods $> 1\%$. $M$: model parameters, $P$: total parameters, $r$: LoRA rank, $Pl$: prompt pool size, $To$: tokens per prompt, $D$: embedding dimension, $SpA$: Sparse Adaptation. Task Parameters and Storage metric are reported in a Per Task manner (except L2P and InfLoRA)

| Method | Task Params | Storage | Acc@top1(%) |
|---|---|---|---|
| ViT-Small (Dosovitskiy et al., 2020) | 0.0 | $M$ | 82.4 |
| RECAST (ours) | $2 \times 10^{-6}P$ | $M$ | 85.0±0.1 |
| L2P (Wang et al., 2022b) | $To * D * M_{pool}$ | $M + Pl$ | 35.9 |
| InfLoRA (Liang & Li, 2024) | $P + 2 \times 10^{-3}P(T-1)$ | $M$ | 88.5 |
| AdaptFormer (Chen et al., 2022) | $6 \times 10^{-4}Pr$ | $M$ | 89.0 |
| MeLo (Zhu et al., 2024) | $8 \times 10^{-4}Pr$ | $M$ | 88.7 |
| DoRA (Liu et al., 2024b) | $1.9 \times 10^{-3}Pr$ | $M$ | 89.3 |
| RoSA (Nikdan et al., 2024)(Full - Att.) | $7 \times 10^{-3}Pr$ | $M$ | 88.6 |
| RoSA (Nikdan et al., 2024)(SpA MLP) | $6 \times 10^{-6}P$ | $M$ | 84.5 |
| RECAST (ours) + Adaptformer | $6 \times 10^{-4}Pr$ | $M$ | **90.1**±0.2 |
| RECAST (ours) + MeLo | $8 \times 10^{-4}Pr$ | $M$ | 89.6±0.2 |
| RECAST (ours) + RoSA(Full - Att.) | $7 \times 10^{-3}Pr$ | $M$ | 88.9 ±0.08 |
| RECAST (ours) + RoSA(SpA MLP) | $6 \times 10^{-6}P$ | $M$ | 85.4±0.05 |

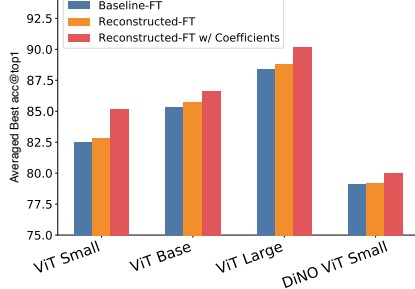

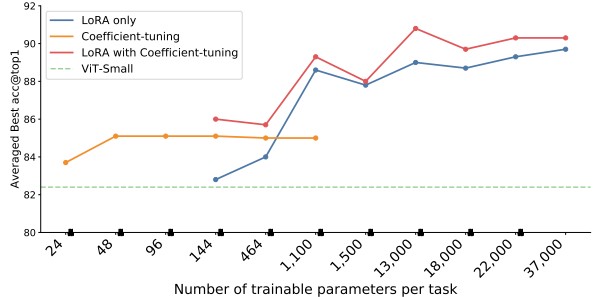

Figure 3: Averaged best Top-1 accuracy across six datasets for ViT models of varying scales, comparing Baseline, RECAST, and RECAST with Coefficient fine-tuning (FT) approaches.

Figure 4: Comparing averaged best Top-1 accuracy across six datasets for ViT-Small across various model configurations. We find RECAST excels in ultra-low parameter ranges (24-96) where LoRA struggles. RECAST w/ LoRA improves performance across all parameter ranges, offering complementary advantages.

$\sim 3.5\%$, adding only $0.0002\%$ task parameters. This represents a significant enhancement with minimal computational overhead. Adapter and reparameterization methods all show strong performance improvements over the baseline: RoSA (Nikdan et al., 2024) (with SpA MLP) improves by $2.5\%$, while InfLoRA (Liang & Li, 2024), MeLo (Zhu et al., 2024), AdaptFormer (Chen et al., 2022), DoRA (Liu et al., 2024b), and RoSA (Full - Att.) all deliver improvements in the $7.5 - 8.5\%$ range. However, combining RECAST with these methods yields additional gains. Both RECAST + RoSA (SpA MLP) and RECAST + AdaptFormer, show a $\sim 1.5\%$ gain over standalone RoSA and AdaptFormer alone, and RECAST + MeLo achieves a $1.0\%$ improvement over MeLo. In contrast, prompting methods like L2P (Wang et al., 2022b), which rely on instance-level prompts, underperform the baseline by $56\%$, struggling to handle diverse domains effectively.

Figure 3 reports the performance of using RECAST over a range of architecture sizes including ViT-Small, ViT-Base, and ViT-Large, demonstrating that our approach generalizes. In addition, we also show that our approach generalizes across pretraining datasets by demonstrating that our approach can boost performance using a DINO Caron et al. (2021) backbone, as well.

**Comparison of Resource Efficiency.** Tables 1 and 2 compare various incremental learning methods for ResNet-34 and ViT-Small backbones, respectively. We see that parameter-free methods (EWC, LWF) maintain minimal storage (only the backbone model parameters) but suffer from per-

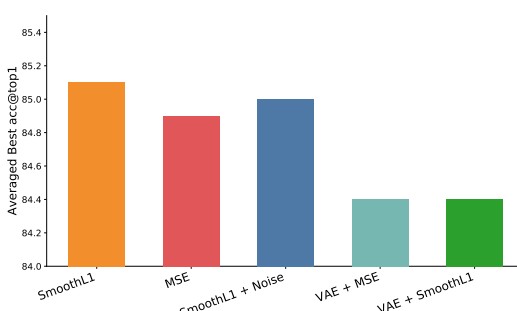

Figure 5: Plot showing average classification accuracy of models reconstructed in different ways. VAE-reconstructed models performed slightly worse than the rest, with Smooth L1 loss providing the best performance

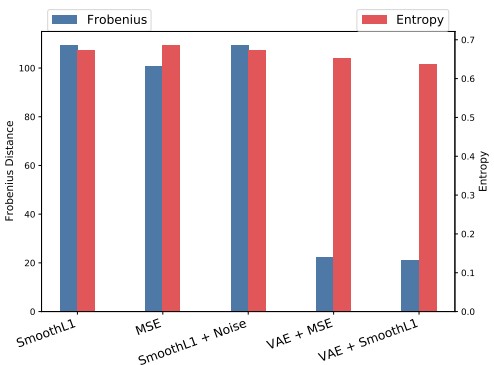

Figure 6: Plot showing Diversity (Frobenius Norm) and adaptability measurements (Entropy) of models reconstructed in different ways. VAE models have shallow template diversity and low internal information content

formance degradation as task diversity increases. Rehearsal-based methods (GDUMB, DER++ & ER-ACE) and HAT offer improved performance at the cost of increased storage requirements, which may limit scalability for long-term learning. Recent parameter-efficient methods balance performance and resource usage. CLR learns 0.3% of backbone parameters, requiring backbone storage plus task-specific depthwise modules. InfLoRA trains the full model for the first dataset and only 0.2% for subsequent tasks. Other LoRA variants train less than 1% per rank. While masking requires training a large number of parameters, both LoRA updates and binary masks (PiggyBack, RoSA Sparse updates) can be mathematically merged to prevent extra storage and communication costs during deployment. Our proposed RECAST method demonstrates unprecedented parameter efficiency, learning only 0.0002% of the total backbone parameters while maintaining only the backbone model in storage. When combined with other efficient methods like MeLo or Adaptformer, RECAST achieves state-of-the-art performance (90.1% Acc@top1 for ViT-Small) with the base storage requirement. This reveals that RECAST not only enhances existing adapter methods but does so in ways that are complementary to their strengths. The impact is delineated clearly in Figure 4, in a range of taskwise parameter counts. It compares three adaptation approaches using ViT-Small backbone (82.4% baseline). Traditional LoRA adaptation (blue) becomes effective only above 464 parameters/task. RECAST (orange) achieves consistent 85% accuracy with just 24-96 parameters, showing marginal improvement beyond this range. The combined RECAST+LoRA (red) peaks at 91% accuracy, outperforming across all parameter ranges, including areas where LoRA alone falters. To scale traditional LoRA implementations to a negligibly small ($< 150$) parameters, we utilized three techniques: (1) lowering LoRA rank, (2) sharing LoRA matrices across layers, and (3) binary masking for pruning. The figure reveals that RECAST can be used effectively in parameter ranges where other reparameterization methods might struggle.

## 4.2 MODEL ANALYSIS

**How does the number of coefficients impact performance?** We analyze the effect of hyperparameters $G$, $n$, and $K$ (Subsection 3.3) through two experiments. First, we vary $n$ while keeping $K = 1$, adjusting $G$ to maintain constant parameters ($G \propto \frac{1}{n}$), comparing three sharing schemes: Low (two-layer sharing), Balanced (four-layer sharing), and Extreme (all-layer sharing). Results are shown in Figure 7. Second, we fix $G$ and $n$ while varying $K$ (since $|C| \propto K$) as depicted in Figure 8. Both experiments use RECAST on MLP and Attention layers of a small ViT model, with marker sizes indicating relative coefficient sizes. Results reveal that adjusting template numbers and sharing schemes has a greater impact than linearly increasing $K$, emphasizing the importance of template diversity and specificity for task adaptation. Additional linear combinations show diminishing returns once sufficient coefficients are available (see the orange line in Figure 4).

**Where do we apply RECAST?** RECAST is applicable to various layers or modules (Appendix A.4.1), with notable insights highlighted in Figures 7 and 8. These figures illustrate the

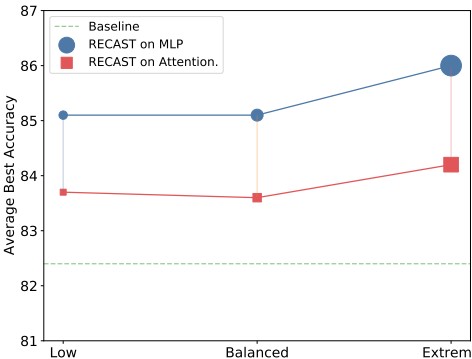

Figure 7: Increasing template numbers (layers in a group) enhances coefficient count and template sharing across the architecture. Low sharing uses G=6, Balanced G=3, and Extreme G=1 (one template bank shared by all network layers). The last setting delivers significant performance improvements.

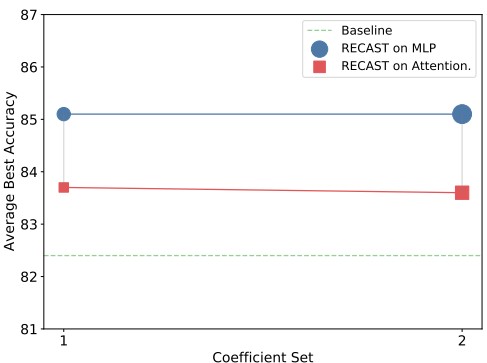

Figure 8: Comparing the strategies of changing coefficients. The size of the markers corresponds to the relative number of coefficients. The impact of increasing coefficients by varying the number of coefficient sets $K$, is much less prominent than changing it through grouping schemes.

differing performance boosts when applying the method to attention layers versus MLP layers. MLP layers, though simpler, have more parameters that may introduce redundancy, making them more amenable to sharing. In contrast, attention layers are complex, involving multiple computational steps to model relationships between input parts rather than directly transforming representations. This complexity leads to a challenging optimization space, hindering effective template sharing and reparameterization. Figure 4 confirms this pattern, showing performance spikes when smaller LoRA matrices target MLP layers (Chen et al., 2022) rather than attention mechanisms (Zhu et al., 2024).

**What is the best way to reconstruct weights?** Section 3.2 described the weight reconstruction methodology for RECAST. Earlier we noted that model expressivity depends more on template bank diversity. To investigate this, we generated five models with the same configurations (6 groups, 2 layers each), varying the reconstruction objective: 1) SmoothL1 loss, 2) MSE loss, 3) SmoothL1 loss with coefficient noise, 4) VAE with MSE loss, and 5) VAE with SmoothL1 loss. All models achieved at least $98.5\%$ reconstruction similarity (Appendix Table 6). Figure 5 shows that SmoothL1 loss (with or without perturbation) provides the most benefit during reparameterization & VAE models provide the least. To understand this, we calculated template diversity within groups using *Frobenius Norm* (diversity metric) and *Entropy* (versatility metric) (See Appendix A.3) Just like their classification performance, SmoothL1 loss shows roughly similar internal patterns with or without noise. The high Frobenius norm for Neural Mimicry (Algorithm 2) with SmoothL1 loss indicates greater template diversity than the rest of the approaches. MSE models showed slightly lower Frobenius norms but higher entropy, suggesting templates are versatile and not overly dependent on dominant components. In contrast, VAEs showed shallow template diversity, limiting adaptability, and lower entropy, indicating reduced versatility. Figure 10 (Appendix A.3) illustrates higher intra-group coefficient similarity in VAE-based models, suggesting multiple layers perform similar operations.

## 5 CONCLUSION

RECAST represents a significant advancement in neural network architecture design, addressing the challenge of optimizing parameter efficiency while maintaining model expressivity. By decomposing network layers into templates and coefficients, it facilitates dynamic, task-specific weight generation with a reduced parameter footprint. Our comprehensive evaluations demonstrate RECAST's efficacy across diverse image classification tasks and vision architectures. Based on our model analysis, future research directions may include enhancing model diversity and coefficient versatility, exploring applications beyond computer vision, and investigating alternative approaches for reconstructing novel parameters to overcome the limitations of existing pretrained weights. Additionally, further research is necessary to elucidate the relationship between model complexity and performance, particularly in the context of model compression techniques.

**Acknowledgments** This material is based upon work supported, in part, by DARPA under agreement number HR00112020054. Any opinions, findings, and conclusions or recommendations expressed in this material are those of the author(s) and do not necessarily reflect the views of the supporting agencies.

## 6 ETHICS STATEMENT

While our work on efficient reparameterization aims to improve model robustness and adaptability, we acknowledge the potential ethical implications of such advancements. First, the incremental learning capabilities enabled by our approach could be beneficial in scenarios like robotics, where systems need to constantly adapt to new categories or environments, as well as provide potential benefits in terms of environmental sustainability by reducing computational footprint. While we intend to democratize access to powerful AI models, we recognize that easier deployment of these models could also lower barriers to potential misuse, such as in surveillance applications that could infringe on privacy rights. Additionally, as with any machine learning model, there are inherent biases and limitations in the training data and model architecture that users should be aware of. As we reconstruct from existing weights, our reconstructed pretrained weights also likely inherits the implicit biases in those existing weights. Thus, we caution against over-reliance on model predictions without human oversight, especially in high-stakes decision-making processes. As researchers, we emphasize the importance of responsible development and deployment of AI technologies, and we encourage ongoing discussions about the ethical use of incremental learning and efficient deep learning models in various applications.

## 7 REPRODUCIBILITY

To ensure the reproducibility of our work, we have taken several key steps. The implementation details of our custom ResNet and Vision Transformer architectures are fully described in the main text, with complete code provided in the supplementary materials. We have also provided the full codes for our framework integrated with other adapter and reparameterization methods we have discussed in the main text. Our reconstruction scripts for both ResNet and ViT models, including the loss functions and training procedures, are also included in the supplementary material. We have specified all hyperparameters, including learning rates, scheduling, and template bank configurations. The base models used for comparison (ResNet34 and ViT-Small) are from widely available libraries (torchvision and timm), ensuring consistent baselines. All experiments were conducted using PyTorch, with specific versions and dependencies listed in the supplementary materials. We have also included our data preprocessing steps and evaluation metrics to facilitate accurate replication of our results. The supplementary material also includes descriptions of how to use the codebase to both reproduce our results, as well as to extend or use by users in a plug-and-play manner.

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

# A  APPENDIX

## A.1  DATASET

In Table 3 we have summarized the class variations and number of samples for the **six** datasets we have used in our TIL experiments. All the datasets were split with $75\%-15\%-15\%$ train-validation-test split. Simple augmentations like resizing, centercropping, horizontal flips, and normalization have been applied for each.

Table 3: Details on the Benchmarking Datasets

| Dataset | Classes | #Sample |
|---|---|---|
| Oxford Flowers (Nilsback & Zisserman, 2008) | 102 | 6553 |
| FGVC Aircrafts (Maji et al., 2013) | 55 | 10001 |
| MIT Scenes (Quattoni & Torralba, 2009) | 67 | 15614 |
| CIFAR100  (Krizhevsky & Hinton, 2009) | 100 | 60000 |
| CIFAR10  (Krizhevsky & Hinton, 2009) | 10 | 60000 |
| CUBs (Wah et al., 2011) | 200 | 11789 |

## A.2  TASKWISE PERFORMANCE BREAKDOWN

**Impact of Task-order**   The sensitivity to task ordering varies significantly across different incremental learning approaches. Methods that rely on knowledge distillation or parameter importance estimation, such as LwF and EWC, exhibit cumulative degradation as knowledge gets diluted across sequential tasks. In particular, LwF's distillation-based approach becomes less effective for later tasks as the knowledge transfer chain grows longer. EWC's reliance on Fisher Information Matrix to protect earlier tasks' parameters can lead to representation bias, where earlier tasks disproportionately influence the learned feature space. Rehearsal-based methods like GDUMB and DER++ face challenges in maintaining balanced representative memory buffers across tasks, though sample selection strategies help mitigate but don't eliminate these order effects. Even masked-based approaches like HAT, despite maintaining unit-wise masks over the entire model, show order sensitivity due to the constrained mask allocation being influenced by earlier tasks. In contrast, both RECAST and several adapter-based baselines (AdaptFormer, MeLo, RoSA, DoRA) maintain absolute parameter inference through strict parameter isolation. As shown in Table 4, these methods achieve consistent performance across tasks, with no significant degradation in earlier task performance as new tasks are learned. For instance, RECAST + AdaptFormer maintains high accuracy across all domains (99.2% on Flowers, 84.7% on Aircraft, 87.1% on Scene), demonstrating that task-specific parameters can effectively adapt while the shared feature space remains stable. This architectural design choice effectively eliminates task interference, as each task's performance remains stable once learned, regardless of the order in which subsequent tasks are introduced. The empirical results suggest that methods employing strict parameter isolation are inherently robust to task ordering effects compared to approaches that modify shared parameters or rely on knowledge transfer between tasks.

## A.3  RECONSTRUCTION ANALYSIS

**Pre-trained Model Details**   We obtained the ViT and DiNO pretrained weights from the official timm repository Wightman (2019). Resnet weights are obtained from Pytorch Hub. The specific models and their corresponding weight URLs are:

- ResNet-34:                    `https://download.pytorch.org/models/resnet34-b627a593.pth`

- ViT-Small:   `https://storage.googleapis.com/vit_models/augreg/S_16-i21k-300ep-lr_0.001-aug_light1-wd_0.03-do_0.0-sd_0.0--imagenet2012-steps_20k-lr_0.03-res_224.npz`

- ViT-Base: `https://storage.googleapis.com/vit_models/augreg/B_16-i21k-300ep-lr_0.001-aug_medium1-wd_0.1-do_0.0-sd_0.0--imagenet2012-steps_20k-lr_0.01-res_224.npz`

- ViT-Large: `https://storage.googleapis.com/vit_models/augreg/L_16-i21k-300ep-lr_0.001-aug_medium1-wd_0.1-do_0.1-sd_0.1--imagenet2012-steps_20k-lr_0.01-res_224.npz`

- DiNO-Small Caron et al. (2021): `https://dl.fbaipublicfiles.com/dino/dino_deitsmall16_pretrain/dino_deitsmall16_pretrain.pth`

Table 4: Model Performance Comparison Across Different Datasets. For InfLoRA the 6 tasks are generated by taking a chunk of 90 classes from the combined dataset - so, the classification result doesn't exactly correspond to each dataset.

| Model Type | Flowers | Aircraft | Scene | CIFAR100 | CIFAR10 | Birds |
|---|---|---|---|---|---|---|
| Small ViT | 97.4 | 66.6 | 82.6 | 72.5 | 94.9 | 81.1 |
| L2P | 0.4 | 5.7 | 0.8 | 56.4 | 75.0 | 77.6 |
| InfLoRA | 99.8 | 89.0 | 88.2 | 84.8 | 84.6 | 84.5 |
| AdaptFormer | 99.2 | 79.6 | 85.7 | 87.6 | 98.1 | 84.3 |
| MeLO | 99.0 | 84.2 | 84.0 | 87.3 | 98.1 | 80.0 |
| Rosa (Sparse MLP 0.001%) | 97.6 | 69.1 | 84.1 | 78.9 | 96.2 | 81.6 |
| RoSA Full | 99.4 | 79.1 | 85.2 | 87.4 | 97.9 | 82.6 |
| DoRA | 99.02 | 86.4 | 84.7 | 87.6 | 98.2 | 80.0 |
| RECAST (Small ViT) | 98.6 | 72.5 | 84.3 | 79.1 | 95.7 | 80.5 |
| RECAST + AdaptFormer | 99.2 | 84.7 | 87.1 | 88.8 | 98.3 | 84.4 |
| RECAST + MeLO | 99.4 | 86.3 | 84.3 | 87.1 | 97.9 | 81.0 |
| RECAST (Sparse MLP 0.001%) | 98.4 | 72.9 | 83.9 | 80.0 | 96.3 | 81.0 |
| RECAST + RoSA Full | 99.2 | 81.9 | 84.7 | 87.5 | 97.9 | 82.4 |
| Resnet34 | 87.6 | 51.9 | 66.9 | 59.8 | 83.0 | 42.0 |
| EWC | 2.6 | 12.1 | 9.0 | 1.7 | 42.2 | 65.4 |
| LWF | 1.4 | 23.1 | 5.6 | 1.7 | 41.1 | 42.9 |
| GDUMB | 38.9 | 25.6 | 33.2 | 4.5 | 69.7 | 36.9 |
| DER++ | 1.3 | 12.5 | 26.1 | 53.9 | 0.7 | 2.5 |
| HAT | 49.1 | 71.9 | 46.5 | 9.9 | 30.6 | 39.8 |
| PiggyBack | 93.9 | 85.4 | 74.4 | 82.6 | 96.6 | 65.0 |
| CLR | 93.9 | 83.3 | 73.3 | 75.9 | 94.3 | 59.0 |
| RECAST (Resnet34) | 89.8 | 52.7 | 65.8 | 61.4 | 83.8 | 44.2 |
| RECAST + PiggyBack | 95.7 | 87.3 | 77.0 | 81.7 | 96.3 | 66.3 |
| RECAST + CLR | 94.7 | 83.7 | 72.1 | 77.4 | 95.6 | 62.1 |

Table 5: Required runtime and hardware utilization to run Neural Mimicry (Section 3.2)

| Model Scale | Peak GPU Memory (GB) | Average CPU Utilization | Per Epoch Processing (ms) | Total Epochs | Wall Clock Time (seconds) |
|---|---|---|---|---|---|
| ViT Small ($\approx 21M$) | 0.3 | 3.7% | 36.2 | 1000 | 37.4 |
| ViT Base ($\approx 86M$) | 1.2 | 3.7% | 112 | 1000 | 102.3 |
| ViT Large ($\approx 304M$) | 4.5 | 3.7% | 340 | 1000 | 245.0 |

**Resource Requirements** The results in Table 5 demonstrate that Neural Mimicry has a modest memory footprint (0.33-4.5GB GPU memory) with minimal CPU overhead ( 3.7%). It also linearly scales with model size. Furthermore, the complete reconstruction is completed in minutes even for large models. Notably, this step achieves 98-100% reconstruction accuracy (Table 6) and requires significantly less resources than model pretraining.

**RECAST is architecture and scale agnostic**   RECAST can reconstruct models across various architecture and scales. In Table 6, we present the results of reconstructed models and the official performance of these models on the Imagenet-1K (Deng et al., 2009) dataset. All reconstructed models in the table have the same configuration - 6 groups, 2 layers in each group and 2 sets of coefficients for each target module (except ViT-Large has 12 groups). The inference comparison on Imagenet tells us that, RECAST can completely emulate the feature generation of any pretrained weight. We also report the averaged best accuracy@top1 across the datasets in figure 3, for the varying scales of the ViT models. In all cases, RECAST perfermos slightly better than baseline without coefficient-tuning, and significantly better with coefficient-tuning.

| Model | Imagenet Acc@top1 | Reconstruction Similarity |
|---|---|---|
| Resnet-34 | 73.3 | - |
| Recon. Resnet-34 | 73.0 | 99% |
| ViT-Small | 74.6 | - |
| Recon. ViT-Small (MLP) | 74.6 | 100% |
| Recon. ViT-Small (Attention) | 74.6 | 100% |
| Recon. ViT-Small (MLP & Attention) | 74.6 | 100% |
| Recon. ViT-Small (VAE + MSE) | 73.8 | 98.5% |
| Recon. ViT-Small (VAE + SmoothL1) | 73.0 | 98.6% |
| ViT-Base | 81.1 | - |
| Recon. ViT-Base | 81.0 | 100% |
| ViT-Large | 84.4 | - |
| Recon. ViT-Large | 84.4 | 100% |
| DINO Small | 77.0 | - |
| Recon. DiNO Small | 76.9 | 99% |

Table 6: Accuracy comparison of reconstructed RECAST models over the ImageNet1k dataset, against the original pretrained models from timm (Wightman, 2019) demonstrates that both backbones are empirically the same. Despite this similarity, RECAST provides reparameterization facilities with $< 0.0002\%$ coefficients

**Template diversity is important for better reparameterization**   In Section A.3, we empirically showed that models with higher Frobenius norm value and higher entropy provides better task-adaptable model. We further breakdown this analysis by first defining the metrics, and then analyzing their layerwise trends for various reconstruction methods.

- **Frobenius Norm:** Calculates the square root of the sum of the absolute squares of its elements. In the context of RECAST, the Frobenius norm is used to measure the diversity among templates within a template bank for each group. For a template bank $\tau_g = \{T_{g,1}, T_{g,2}, \ldots, T_{g,n}\}$ in group $g$, the average Frobenius norm can be calculated as: $\text{avg\_frobenius}_g = \frac{1}{n(n-1)} \sum_{i=1}^n \sum_{j=i+1}^n \|T_{g,i} - T_{g,j}\|_F$. Squaring the differences makes the metric more sensitive to large deviations between templates. This property helps identify when templates are significantly different, not just slightly varied.

- **Entropy** Quantifies the average amount of information contained in a set of values. For each template bank $\tau_g$, we calculate the entropy of singular values to measure the balance of importance among the components of the templates. The average entropy for the template bank in group $g$ is calculated as: $\text{avg\_entropy}_g = \frac{1}{n} \sum_{i=1}^n H(T_{g(l),i})$, where $H(T_{g,i})$ is the entropy for a single template $T_{g,i}$

The Frobenius norm helps ensure diversity among $T_{g,i}$, allowing for a wide range of possible $W_{l,m}$. A higher score means more difference among the template. The entropy ensures that each $T_{g,i}$ is itself balanced and flexible, contributing to the adaptability of the reconstructed weights. A higher entropy corresponds to better information quality within each group. Together, these metrics provide a comprehensive view of the template characteristics in RECAST, helping to optimize both the diversity of the template bank and the flexibility of individual templates for efficient and effective weight reconstruction. In Figure 9a and 9b, we show the change of Frobenius norm and

Entropy. The VAE models perform very low in terms of the Frobenius metric, although their pattern varies depending on the loss function. This is also true for the models generated through Neural Mimicry 3.2,*i.e.* models reconstructed with MSE loss and SmoothL1 loss performance almost similar on classification task but differs in entropy pattern and Frobenius norm. Another observation is that noise perturbations don't seem to impact these two metrics at all.

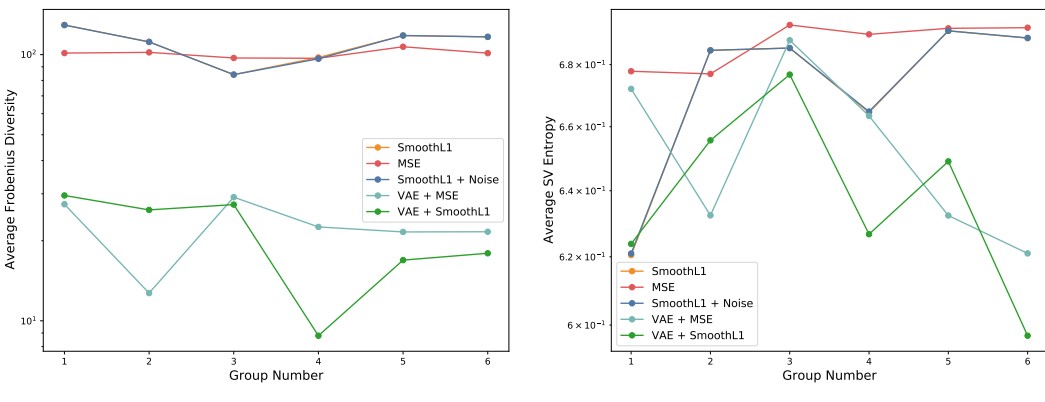

(a) Change of Frobenius norm across the groups      (b) Change of Entropy across the groups

Figure 9: Analyzing the strategies of changing reconstruction schemes by quantifying their respective Frobenius Norm and Entropy metrics (as defined in A.3) across 6 groups of layers of a ViT. It can be observed that the VAE-reconstructed models generally demonstrate significantly low Group-wise Frobenius score and comparatively lower Entropy.

We define another metric the **intra-group coefficient similarity** to measure how similar the coefficient vectors are between different layers within the same group of RECAST model. For layers $l1$ and $l2$ in the same group: $sim(l1, l2) = cosine\_similarity(C_{l1}, C_{l2})$ Where $C_l$ is the flattened and concatenated vector of all coefficients for layer l. If similarities are generally low within a group, it means each coefficient is using the shared templates in unique ways. This could indicate that each layer is capturing different features or transformations. The Figure 10, shows that the later groups in the VAE models share very high intra-group coefficient similarity indicating that layers within each group are using templates very similarly or there are redundancies in the network.

## A.4 RECAST ADAPTATION

### A.4.1 RECAST-ADAPTED COMPONENTS

Our framework can be adapted to various neural network components. Below we provide the details of how they may be implemented in Neural Networks.

**Fully-connected Layers** We use RECAST here to generate the weights for the full-connected layers of a network. Here, $T_i \in \mathbb{R}^{d_{out} \times d_{in}}$. $d_{out}$ is the number of output features and $d_{in}$ is the number of input features. Coefficient shape, $C_i^j \in \mathbb{R}^{1 \times 1}$, represents a scalar value for each template with additional broadcasting dimensions. After generating final weight using Eq. 2, it's used as $Y = f(X \cdot W_{final} + b)$, where $X \in \mathbb{R}^{batch\_size \times d\_in}$ and $Y \in \mathbb{R}^{batch\_size \times d\_out}$

**Attention QKV Matrices** In attention mechanisms, RECAST generates a single matrix for the combined query, key, and value projections. This uses a template shape $T_i \in \mathbb{R}^{3d \times d}$, where $d$ is the embedding dimension and coefficient shape, $C_i^j \in \mathbb{R}^{1 \times 1 \times 1}$. The weight generation process is similar to the FC layer case. The resulting $W_{final}$ is then used to compute $Q$, $K$, and $V$. For input shape $X \in \mathbb{R}^{batch\_size \times seq\_length \times d}$ we can obtain $Q, K, V \in \mathbb{R}^{batch\_size \times seq\_length \times d}$

**Convolution Kernels** For convolutional layers, RECAST generates filter kernels that are applied across the input feature maps. Here, $T_i \in \mathbb{R}^{c_{out} \times c_{in} \times K \times K}$. $c_{out}$ is the number of output channels, $c_{in}$ is the number of input channels, and $K$ is the kernel size. The weight generation process follows

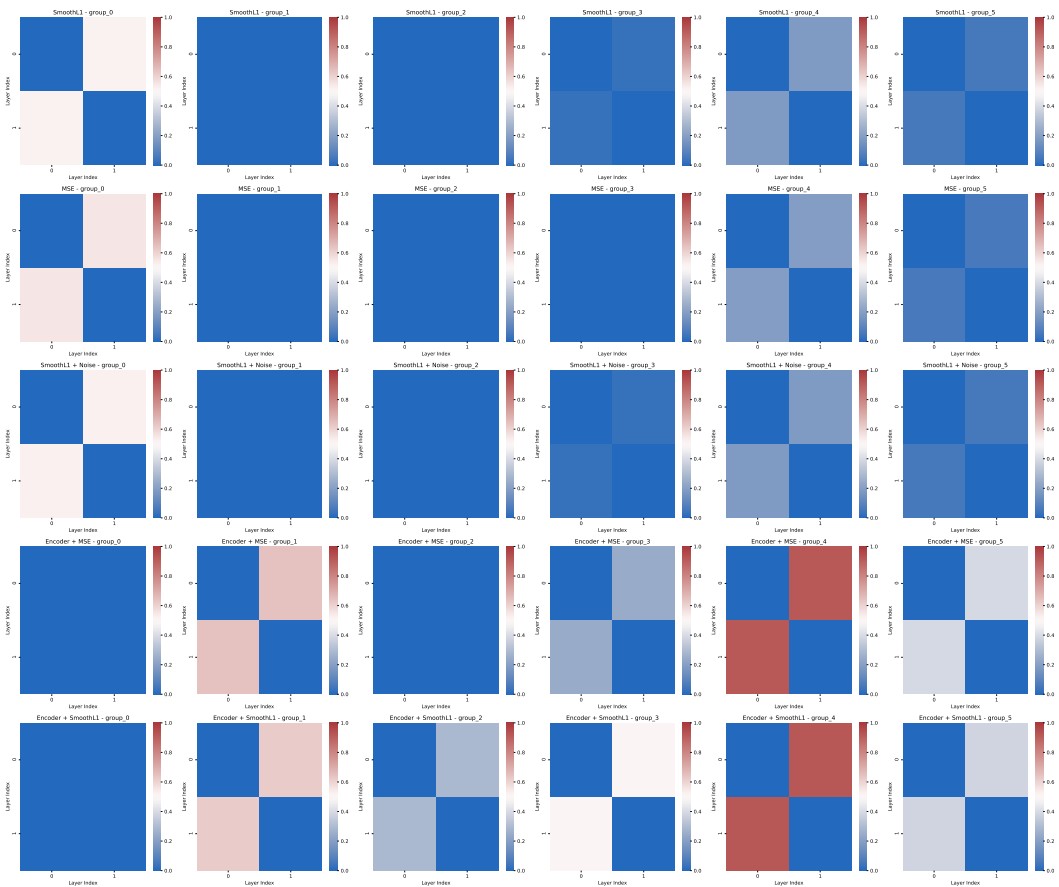

Figure 10: Intra-group Coefficient Similarity across the groups of similar configuration models, that have been reconstructed in different ways. The later groups in the VAE/Encoder models share very high intra-group coefficient similarity and that layers within each group are using templates very similarly or there are redundancies in the network.

the same pattern as before. The resulting $W_{final}$ is then used as the convolution kernel:
$Y = conv2d(X, W_{final})$, where $X \in \mathbb{R}^{batch\_size \times c\_in \times h\_in \times w\_in}$ and $Y \in \mathbb{R}^{batch\_size \times c\_out \times h\_out \times w\_out}$

### A.4.2 RECAST INTEGRATION WITH EXISTING METHODS

RECAST complements existing incremental learning methods as a task-adaptable weight generation backbone. For CNN architectures, RECAST generates base weights that PiggyBack's (Mallya et al., 2018) binary masks can modulate and provides primary convolution weights that CLR's Ge et al. (2023) Ghost modules can refine. In transformer architectures, RECAST produces core weight matrices that methods like MeLO, AdaptFormer (for query/value and MLP components), DoRA (for magnitude-direction decomposition), and RoSA (for sparse adaptations) can then modify.

For example, in standard LoRA, a weight matrix $W \in \mathbb{R}^{d_{out} \times d_{in}}$ is modified by adding a low-rank update:

$$W_{\text{LoRA}} = W_0 + BA \tag{4}$$

where $B \in \mathbb{R}^{d_{out} \times r}$, $A \in \mathbb{R}^{r \times d_{in}}$, and $r$ is the rank.

When integrating with RECAST, instead of using a fixed $W_0$, we use the dynamically generated weight matrix from RECAST. From Equation 1

$$W_{\text{RECAST}} = \frac{1}{K} \sum_{k=1}^{K} \sum_{i=1}^{n} T_{g,i} \cdot C_{l,m,i}^{k} \tag{5}$$

where $T_{g,i}$ are the templates and $C_{l,m,i}^k$ are the coefficients. The final integrated approach combines both methods:

$$W_{\text{combined}} = W_{\text{RECAST}} + BA \tag{6}$$

The total trainable parameters for the combined approach are:

- RECAST: $O(nK)$ coefficients per module
- LoRA: $O(r(d_{in} + d_{out}))$ parameters per module

where $n$ is the number of templates, $K$ is the number of coefficient sets, and $r$ is the LoRA rank.

Similar principles apply when integrating with other methods:

**CLR (Channel-wise Lightweight Reprogramming) Ge et al. (2023):**

$$Y = \text{Ghost}(\text{Conv}(X, W_{\text{RECAST}})) \tag{7}$$

where $\text{Ghost}(\cdot)$ represents the lightweight channel-wise transformations through depthwise-separable convolutions.

**PiggyBack (Mallya et al., 2018):**

$$W_{\text{combined}} = W_{\text{RECAST}} \odot M \tag{8}$$

where $M$ is a binary mask matrix.

**DoRA (Liu et al., 2024b):**

$$W_{\text{combined}} = \|W_{\text{RECAST}}\| \cdot \frac{W_{\text{RECAST}} + BA}{\|W_{\text{RECAST}} + BA\|} \tag{9}$$

where $\| \cdot \|$ denotes the magnitude and $BA$ is the LoRA update.

**RoSA (Nikdan et al., 2024):**

$$W_{\text{combined}} = W_{\text{RECAST}} + S \odot W_{\text{RECAST}} + (BA) \tag{10}$$

where $S$ is a sparse binary mask applied to the LoRA update.

