# OpenReview forum: "RECAST: Reparameterized, Compact weight Adaptation for Sequential Tasks"
_ICLR.cc/2025/Conference — ICLR 2025 Poster_

### Official Review · Reviewer_uU62 · 2024-10-31

**Soundness:** 2
**Presentation:** 2
**Contribution:** 3
**Rating:** 6
**Confidence:** 3

**Summary:**

RECAST is a Task Incremental Learning (TIL) method that efficiently reparameterizes model components to emulate a pretrained target network by decomposing weights into shared templates and module-specific coefficients. The shared templates, stored in a template bank, match the shape of each layer and remain fixed across tasks, while the module-specific coefficients are fine-tuned for each new task.

Through a process called Neural Mimicry, RECAST selects a target model and combines the templates and coefficients to approximate it. The approach is architecture-agnostic and requires minimal parameter updates across tasks.

**Strengths:**

- Clear presentation of results, focusing on both parameter efficiency and storage alongside accuracy.
- The method enhances existing adapter-based methods, making it compatible and valuable as a supplementary tool.
- It is versatile across architectures, with applicability to both CNNs and Transformers.
- The setting is flexible, allowing adjustment of groups *G* and templates *n* to achieve the desired level of model compression.

**Weaknesses:**

I recommend adding another baseline based on rehearsal, such as ER[1], ER-ACE[2], or DER++[3]. While the proposed method is more aligned with adapter-based approaches, claiming strong performance in settings utilizing a memory buffer without including one of these established baselines for comparison seems somewhat unfair.

I understand that the accuracy reported is averaged across the six datasets used at the end of training. However, I'm curious if accuracy varies between tasks encountered earlier in training and those seen later. Including a table that shows accuracy results for individual tasks would strengthen the paper’s validity. This would also relate to Question 1 regarding the influence of task order.


    [1] Jeffrey S Vitter. Random sampling with a reservoir. ACM Transactions on Mathematical Software (TOMS), 11(1):37–57, 1985
    [2] Lucas Caccia, Rahaf Aljundi, Nader Asadi, Tinne Tuytelaars, Joelle Pineau, and Eugene Belilovsky. New Insights on Reducing Abrupt Representation Change in Online Continual Learning. In International Conference on Learning Representations Workshop, 2022.
    [3] Pietro Buzzega, Matteo Boschini, Angelo Porrello, Davide Abati, and Simone Calderara. Dark Experience for General Continual Learning: a Strong, Simple Baseline. In Advances in Neural Information Processing Systems, 2020.

**Questions:**

1.  Does the task order adhere to the sequence in which the datasets are presented in the text (lines 294-296), or does averaging across three different runs involve altering the task order?
I believe exploring the latter could be an intriguing study on how task order impacts weight adaptation and whether the learned coefficients are affected by such variations.

**About LoRA and Figure 3**

2. It’s not entirely clear what is being evaluated in Figure 3.
While a comparison with LoRA is certainly necessary in this work, could you clarify what "CF" stands for in the figure?

3. Additionally, rather than comparing to LoRA-only, it might be more informative to use the same baseline using LoRA in one case and RECAST in the other, e.g., fine-tuning the same backbone with the two methods.
It’s possible that Figure 3 already presents this setup, as it seems ViT-Small is used as the backbone and lower bound. If this is indeed the case, the figure caption and related section would benefit from clarification, with additional emphasis on these details to enhance interpretability.

4.  Furthermore, the comparison does not appear to fully explore the parameter range for both methods; although the x-axis in the plot represents the number of parameters, the methods are not evaluated across the entire range, and a comparison with the same number of parameters is only partially explored. Could you clarify the reasoning behind this?

**CL Setting**
5. Is there a specific reason why the authors have chosen to focus exclusively on the Task Incremental Setting, rather than exploring the more challenging Class Incremental Setting?

**Details Of Ethics Concerns:**

No ethics concerns.

---

> ### Author Response · Authors · 2024-11-23
> **To Reviewer uU62**
>
> We deeply appreciate the reviewers effort and providing us the opportunity to elaborate on several important aspects of our work. Below we provide our responses by breaking it down by the points presented by the reviewer.
>
> > **W1** I recommend adding another baseline based on rehearsal, such as ER[1], ER-ACE[2], or DER++[3]. While the proposed method is more aligned with adapter-based approaches, claiming strong performance in settings utilizing a memory buffer without including one of these established baselines for comparison seems somewhat unfair
>
> Thank you for your suggestions to add more established baselines. Following your recommendations, we have run experiments on both DER++ and ER-ACE, as well as reported their performances in Table 1.  We have used an existing repository[1] for both implementations but had to make minor changes in the dataloading and sampling step to accommodate our target datasets. Below we attached a snapshot of their performance, compared to  RECAST-Resnet. These new rehearsal methods also perform noticeably poorly compared to our proposed method.
> | Method | Learnt Params | Storage | Acc@top1(%) |
> |---------------------------------|---------------------|-------------|-------------------|
> | ER-ACE | P | M + R | 36.1 |
> | DER++ | P | M + R | 16.2 |
> | RECAST-ResNet34 | 3 * 10^{-6}P |  M | 66.3 +- 0.05 |
>
>
> ```
> [1] pclucas14. “GitHub - Pclucas14/AML.” GitHub, 2020, github.com/pclucas14/AML. Accessed 22 Nov. 2024.
> ```
> > **W2**  I understand that the accuracy reported is averaged across the six datasets used at the end of training. However, I'm curious if accuracy varies between tasks encountered earlier in training and those seen later. Including a table that shows accuracy results for individual tasks would strengthen the paper’s validity. This would also relate to Question 1 regarding the influence of task order.
>
> > **Q1** Does the task order adhere to the sequence in which the datasets are presented in the text (lines 294-296), or does averaging across three different runs involve altering the task order?I believe exploring the latter could be an intriguing study on how task order impacts weight adaptation and whether the learned coefficients are affected by such variations.
>
> The task order for the reported  experiments is [‘Oxford Flowers’, ‘FGVC Aircrafst’, ‘MIT Scenes’, ‘CIFAR100’, ‘CIFAR10’, ‘CUBS’]. We have updated the dataset details Table 4 in the Appendix to reflect this ordering. We do not change this order while running the experiments across different seeds.
>
> Task-order is not applicable in our particular TIL setting and for most of the baselines we have reported, including our approach.  Some baselines like LwF and EWC are impacted by task-ordering due to cumulative forgetting and representation bias. Rehearsal-based methods are also affected by task order, though sample selection can mitigate but not eliminate the order effect.HAT, which maintains a unitwise mask, gets impacted by order since earlier tasks influence the limited mask allocation. However, all the other majority of the baselines maintain complete parameter-isolation, i.e: no task-interference making task-ordering less important for the setting. We appreciate the chance to elaborate on this and have added a more in-depth discussion in  Appendix Section A.2
> Additionally, a new Table 4 has been added to the appendix to give a task wise performance across the baseline.
> Here’s a partial snapshot of the table 4 of the Appendix :
>
> | Model Type                  | Flowers | Aircraft | Scene | CIFAR100 | CIFAR10 | Birds | Average |
> |-----------------------------|---------|----------|-------|----------|---------|-------|---------|
> | Small ViT                   | 97.4    | 66.6     | 82.6  | 72.5     | 94.9   | 81.1  | 82.4   |
> | L2P                         | 0.4     | 5.7      | 0.8   | 56.4     | 75.0    | 77.6  | 35.9  |
> | InfLoRA                     | 99.9   | 89.1    | 88.3 | 84.9    | 84.7   | 84.5  | 88.5   |
> | AdaptFormer                 | 99.2    | 79.6     | 85.7  | 87.6     | 98.1    | 84.3  | 89.0   |
> | MeLO                        | 99.0    | 84.2     | 84.0  | 87.3     | 98.1    | 80.0  | 88.7   |
> | RoSA Full                   | 99.4    | 79.1     | 85.2  | 87.4     | 97.9    | 82.6  | 88.6   |
> | DoRA                        | 99.0   | 86.4     | 84.7  | 87.6     | 98.2    | 80.0  | 89.3   |
> | RECAST (Small ViT)          | 98.6    | 72.5     | 84.3  | 79.1     | 95.7    | 80.5  | 85.1   |
> | RECAST + AdaptFormer        | 99.2    | 84.7     | 87.1  | 88.8     | 98.3    | 84.4  | 90.4  |
> | RECAST + MeLO               | 99.4    | 86.3     | 84.3  | 87.1     | 97.9    | 81.0  | 89.4   |
> | RECAST + RoSA Full          | 99.2    | 81.9     | 84.7  | 87.5     | 97.9    | 82.4  | 88.9  |
>
> As can be seen above, per-task results generally follow averaged results, with the ranking of methods for each task being largely (but not entirely) consistent with overall results.

---

> ### Author Response · Authors · 2024-11-23
> **Contd. To Reviewer uU62**
>
> >**Q2** It’s not entirely clear what is being evaluated in Figure 3. While a comparison with LoRA is certainly necessary in this work, could you clarify what "CF" stands for in the figure?
>
> Thank you for the opportunity to clarify on this: The figure 3 demonstrates a performance comparison across different parameter counts between three approaches: 1. LoRA implementations, 2. RECAST's coefficient-tuning mechanism, and 3. The combination of both approaches.
>
> You are correct that "CF" in the figure legend should be explicitly labeled as "Coefficient-tuning" to maintain consistency with the terminology used throughout the paper. We have updated the figure and caption accordingly.
> The key finding demonstrated in Figure 3 remains as before - RECAST's coefficient-tuning approach provides substantial performance benefits in extremely parameter-constrained settings where traditional LoRA implementations struggle to operate effectively. Furthermore, when combined with LoRA, our approach consistently enhances the baseline LoRA performance across all parameter ranges tested.
>
> > **Q3** Additionally, rather than comparing to LoRA-only, it might be more informative to use the same baseline using LoRA in one case and RECAST in the other, e.g., fine-tuning the same backbone with the two methods. It’s possible that Figure 3 already presents this setup, as it seems ViT-Small is used as the backbone and lower bound. If this is indeed the case, the figure caption and related section would benefit from clarification, with additional emphasis on these details to enhance interpretability.
>
> We confirm that your interpretation is correct. All reported transformer models used Small-ViT as the backbone (RECAST models use equivalent RECAST-ViT) and similarly, all CNN-based methods use ResNet-34 backbone.
> Figure 3 demonstrates performance comparisons, by first showing the performance of the common backbone (green dashed line at 82.4% accuracy). The other lines represent three adaptation approaches of this backbone:
> 1. LoRA adaptation (blue line): Shows performance scaling with parameter count, becoming effective only above ~464 parameters per task. This represents the traditional low-rank adaptation approach.
> 2. RECAST coefficient-tuning (orange line): Demonstrates consistent performance improvement (~85%) even with extremely few parameters (24-96). The flat performance curve indicates that additional coefficients provide diminishing returns, which is why we focus on the lower parameter range where RECAST shows its primary advantage.
> 3. Integrated approach (red line): Combines RECAST's coefficient-tuning with LoRA, showing complementary benefits across all parameter ranges. Notably, this approach maintains superior performance even in low-parameter regions where LoRA alone struggles, achieving peak accuracy of ~91%.
> To accomodate your suggestions about clarity, we have also updated the caption of the figure 3. Additionally,  we have also added a paragraph in Section 3.1 to discuss the figure in more detail.
>
> >  **Q4** Furthermore, the comparison does not appear to fully explore the parameter range for both methods; although the x-axis in the plot represents the number of parameters, the methods are not evaluated across the entire range, and a comparison with the same number of parameters is only partially explored. Could you clarify the reasoning behind this?
>
> RECAST cannot be meaningfully evaluated beyond 96 parameters because our experiments show no performance gains with additional parameters (see flat orange line in Figure 3). LoRA cannot be evaluated below 144 parameters because that is the lowest number of parameters it can achieve even with extreme optimization (reduced rank, shared matrices, 98% sparsity).

---

> ### Author Response · Authors · 2024-11-23
> **Contd.(2) To Reviewer uU62**
>
> > **Q5** Is there a specific reason why the authors have chosen to focus exclusively on the Task Incremental Setting, rather than exploring the more challenging Class Incremental Setting?
>
> We focused on Task Incremental Learning (TIL) following standard evaluation protocols from prior work [1,2,3,4] to enable direct comparisons with existing methods. Moreover, as our primary contribution is demonstrating extreme parameter efficiency (fewer than 50 parameters per task, orders of magnitude less than competing methods), TIL provides a clear and controlled setting to validate this claim.  At the same time, TIL allows us to demonstrate the utility of using RECAST as a reparameterization framework.
>
> Several prior works, including LoRA variants, have explored how TIL methods can be adapted to CIL settings [5,6,7]. As we show RECAST can seamlessly integrate with existing TIL methods, there is no technical limitation preventing RECAST from working in Class Incremental Learning (CIL).
> ```
> [1] Yunhao Ge, Yuecheng Li, Shuo Ni, Jiaping Zhao, Ming-Hsuan Yang, and Laurent Itti. Clr: Channel-wise lightweight reprogramming for continual learning, 2023
> [2]  Mitchell Wortsman, Vivek Ramanujan, Rosanne Liu, Aniruddha Kembhavi, Mohammad Rastegari, Jason Yosinski, and Ali Farhadi. Supermasks in superposition. Advances in Neural Information Processing Systems, pages 1517315184, 2020
> [3] Brian Cheung, Alex Terekhov, Yubei Chen, Pulkit Agrawal, and Bruno Olshausen. Superposition of many models into one. arXiv preprint arXiv:1902.05522, 2019
> [4] Arun Mallya and Svetlana Lazebnik. Piggyback: Adding multiple tasks to a single, fixed network by learning to mask. CoRR, abs/1801.06519, 201
> [5] D. Abati, J. Tomczak, T. Blankevoort, S. Calderara, R. Cucchiara, and B. E. Bejnordi, “Conditional channel gated networks for task-aware continual learning,” in Proceedings of the IEEE Conference on Computer Vision and Pattern Recognition, 2020, pp. 3931–3940
> [6] C. Henning, M. Cervera, F. D’Angelo, J. Von Oswald, R. Traber, B. Ehret, S. Kobayashi, B. F. Grewe, and J. Sacramento, “Posterior meta-replay for continual learning,” Advances in Neural Information Processing Systems, vol. 34, pp. 14135–14149, 2021
> [7] Guo, Haiyang et al. “PILoRA: Prototype Guided Incremental LoRA for Federated Class-Incremental Learning.” (2024).
> ```

---

> > ### Comment · Reviewer_uU62 · 2024-11-25
> >
> > I would like to thank the reviewers and sincerely appreciate the time they dedicated to providing their responses. Taking into consideration the revision and enhancement efforts applied to the main paper based on both my review and those of others, I will increase my score.

---

> > > ### Author Response · Authors · 2024-12-03
> > > **Official response by Authors**
> > >
> > > Thank you so much for taking the time to review the updated work, and increasing your scores. Your reviews have truly helped us enhance the quality of our research.

---

### Official Review · Reviewer_YqZe · 2024-11-04

**Soundness:** 3
**Presentation:** 1
**Contribution:** 2
**Rating:** 6
**Confidence:** 4

**Summary:**

In this work, the authors propose a parameter efficient task incremental learning method. Specifically, the proposed method reparameterizes the original model by using pre-trained template banks and lightweight learnable coefficients for each tasks during incremental learning. In the experimental results, the proposed method shows better accuracy with less trainable parameters when combining with previous incremental learning methods.

**Strengths:**

1. The proposed method can reduce learnable parameter for incremental learning.

2. In the experiments, the proposed methods shows better accuracy with less trainable parameters when combining with previous incremental learning methods.

**Weaknesses:**

1. The paper is not well written and not well organized. For example:

    (1) The proposed method and related works are mixed together in Section 2.

    (2) Subsection 2.1 introduces too many concepts from later sections, making it difficult to follow.

    (3) The opening paragraphs of Subsections 2.1 and 2.2 are largely redundant.

     (4) The overview of the proposed RECAST method (Fig. 2) should be introduced at the beginning of Section 2, but it isn’t presented until Subsection 2.2.

2. The proposed RECAST method itself does not seem directly related to task incremental learning.

3. The paper lacks sufficient background information, especially on relevant works like Template Mixing methods, making it difficult to assess the novelty of the approach.

4. It is unclear how RECAST integrates with previous incremental learning methods, as shown in Tables 1 and 2.

**Questions:**

Overall, this work appears to introduce a new parameter-efficient task incremental learning method, which significantly reduces the number of learnable parameters while improving accuracy when combined with previous incremental learning techniques. However, the current version has presentation quality issues, and the contribution of the technical aspects remains unclear. My specific questions are as follows:

1. What are the technical differences between the proposed RECAST and previous Template Mixing methods? For example:
   - How does the design of template banks and coefficients differ in RECAST?
   - How does RECAST support incremental learning with substantially fewer learnable parameters, whereas Template Mixing methods do not achieve this?

   It's essential to clarify the specific differences to highlight the novelty of the proposed method.

2. How does RECAST balance model plasticity and stability in incremental learning? From my understanding, RECAST reparameterizes the model using lightweight coefficients and frozen template banks for each downstream task, making it a parameter-efficient fine-tuning method rather than one specifically designed for incremental learning. In the experiments, RECAST is combined with existing incremental learning methods to achieve better performance. If this understanding is correct, it would be useful to compare RECAST with other parameter-efficient fine-tuning methods, such as prompt-tuning, LoRA, and DoRA.

3. How does RECAST integrate into existing incremental learning methods, as shown in Tables 1 and 2?

**Details Of Ethics Concerns:**

No ethics concerns found.

---

> ### Author Response · Authors · 2024-11-23
> **To Reviewer YqZe**
>
> We deeply appreciate the reviewers effort and providing us the opportunity to elaborate on several important aspects of our work. Below we provide our responses by breaking it down by the points presented by the reviewer.
>
> > **W1** The paper is not well written and not well organized. For example:
>
> > (1) The proposed method and related works are mixed together in Section 2.
>
> > (2) Subsection 2.1 introduces too many concepts from later sections, making it difficult to follow.
>
> > (3) The opening paragraphs of Subsections 2.1 and 2.2 are largely redundant.
>
> > (4) The overview of the proposed RECAST method (Fig. 2) should be introduced at the beginning of Section 2, but it isn’t presented until Subsection 2.2.
>
>  We sincerely appreciate these insightful suggestions for improving the paper's structure. Our original organization aimed to contextualize each component of RECAST alongside relevant prior work, helping readers understand our contributions with existing methods. We agree that the presentation can be enhanced for clarity.
> Following your feedback, we made the following structural improvements:
>
>  (1) Moved high-level concepts, and related studies to the overview part of Section 2
>
>  (2) Changed the order of the component-specific subsections. New subsection 2.1 is now Weight Decomposition and Subsection 2.2 is Neural Mimicry
>
>  (3) Removed the redundant paragraphs from the component-specific subsections
>
>  (4) Moved Figure 2 to Section 2's introduction to provide readers with a clear, high-level overview of RECAST before diving into technical details.
>
> > **W2** The proposed RECAST method itself does not seem directly related to task incremental learning.
>
>  While RECAST was evaluated on task incremental learning, its core innovations - efficient weight generation through template banks and Neural Mimicry reconstruction - are general techniques that can benefit many scenarios requiring parameter-efficient model adaptation. Neural Mimicry can convert any pretrained model (CNNs, Transformers, etc.) into a more parameter-efficient form while preserving performance. On the other hand, the weight decomposed architecture allows easier model updating in distributed systems by only sending small coefficient updates. These components could be particularly valuable in edge computing, and federated learning scenarios where communication cost and update efficiency are critical, which highlight potential benefits and impacts of our work.
> Our experiments are set up similarly to other TIL studies like CLR[1], SUPSUP[2], PSP[3], Piggybank[4] etc., and we establish the proposed method’s superiority in this setting by:
>  -  Preventing catastrophic forgetting through task-specific coefficients while keeping templates fixed
>  -  Requiring minimal parameters per task (<50) compared to traditional approaches
>  -  Enabling seamless task switching by simply changing coefficient sets
>  -  Maintaining consistent performance across sequential tasks
> ```
> [1] Yunhao Ge, Yuecheng Li, Shuo Ni, Jiaping Zhao, Ming-Hsuan Yang, and Laurent Itti. Clr: Channel-wise lightweight reprogramming for continual learning, 2023
> [2]  Mitchell Wortsman, Vivek Ramanujan, Rosanne Liu, Aniruddha Kembhavi, Mohammad Rastegari, Jason Yosinski, and Ali Farhadi. Supermasks in superposition. Advances in Neural Information Processing Systems, pages 1517315184, 2020
> [3] Brian Cheung, Alex Terekhov, Yubei Chen, Pulkit Agrawal, and Bruno Olshausen. Superposition of many models into one. arXiv preprint arXiv:1902.05522, 2019
> [4] Arun Mallya and Svetlana Lazebnik. Piggyback: Adding multiple tasks to a single, fixed network by learning to mask. CoRR, abs/1801.06519, 201
> ```

---

> ### Author Response · Authors · 2024-11-23
> **Contd. To Reviewer YqZe**
>
> > **W3** The paper lacks sufficient background information, especially on relevant works like Template Mixing methods, making it difficult to assess the novelty of the approach.
>
> > **Q1**  What are the technical differences between the proposed RECAST and previous Template Mixing methods?
>
> We added a more comprehensive discussion on this at the beginning of Section 2. We provide the answers to your specific questions below
>
> > **Q1.1** How does the design of template banks and coefficients differ in RECAST?
>
>  In principle, our work is similar to the works by Savarese & Maire[1] and Plummer et al.[2]. Savarese & Maire et al. applied the concept of basis matrices and scalar coefficients to create a hybrid recurrent CNN architecture. Plummer et al. experimented with this concept to learn where and how to share parameters in a network to fit any given parameter budget. In both cases, the models need to be trained from scratch to perform their targeted task. Our approach, on the other hand, focuses on extremely lightweight model reparameterization. RECAST introduces a multi-set coefficient mechanism (Section 2.1) that enables richer weight generation than single-coefficient Template Mixing methods. This increases expressivity without significant parameter overhead. Each module can generate nᴷ distinct weight configurations while only requiring n×K parameters . This is further demonstrated in  Figure 6 of main paper and discussed in section 3.2.
>
> Similarly, RECAST overcomes the resource limitations of the prior techniques by introducing Neural Mimicry - an efficient novel reconstruction pipeline that enables effective template bank learning without expensive full model retraining. The efficiency of Neural Mimicry is now included in the Appendix Section A.3 Table 5 (as well as provided below)
> | Model (size)      | Peak GPU | CPU Usage | Per Epoch | Total Time |
> | ----------------- | -------- | --------- | --------- | ---------- |
> | ViT-Small (~21M)  | 0.3GB   | 3.7%     | 36.2ms    | 37.4s     |
> | ViT-Base (~86M)   | 1.2GB    | 3.7%     | 112ms     | 102.3s     |
> | ViT-Large (~307M) | 4.5GB    | 3.7%     | 340ms     | 245.0s     |
>
> The results demonstrate that Neural Mimicry has a modest memory footprint (0.33-4.5GB GPU memory) with minimal CPU overhead (~3.7%).
> ```
> [1] Pedro Savarese and Michael Maire. Learning implicitly recurrent cnns through parameter sharing,2019.
> [2] Bryan A. Plummer, Nikoli Dryden, Julius Frost, Torsten Hoefler, and Kate Saenko. Neural parameter allocation search, 2022
> ```
>
> > **Q1.2** How does RECAST support incremental learning with substantially fewer learnable parameters, whereas Template Mixing methods do not achieve this?
>
>  As discussed in the previous response, existing methods that utilize the concept template mixing in any capacity - require full training from scratch for any downstream task. Our Neural Mimicry pipeline (Section 2.2) enables direct reconstruction of pre-trained weights,  allowing RECAST to leverage pretrained model knowledge while only requiring a small number of tunable parameters. The reconstruction process (Algorithm 2) ensures high fidelity (98-100% reconstruction similarity, Table 6, reproduced below).
>
> | Model                             | Imagenet   | Reconstruction |
> |-----------------------------------|------------|----------------|
> |                                   | Acc@top1   | Similarity     |
> | Resnet-34                         | 73.3       | -              |
> | Recon. Resnet-34                  | 73.0       | 99\%           |
> | ViT-Small                         | 74.6       | -              |
> | Recon. ViT-Small (MLP)            | 74.6       | 100\%          |
> | Recon. ViT-Small (VAE + MSE)      | 73.8       | 98.5\%         |
> | Recon. ViT-Small (VAE + SmoothL1) | 73.0       | 98.6\%         |
> | ViT-Base                          | 81.1       | -              |
> | Recon. ViT-Base                   | 81.0       | 100\%          |
> | ViT-Large                         | 84.4       | -              |
> | Recon. ViT-Large                  | 84.4       | 100\%          |
> | DINO Small                        | 77.0       | -              |
> | Recon. DiNO Small                 | 76.9       | 99\%           |

---

> ### Author Response · Authors · 2024-11-23
> **Contd.(2) To Reviewer YqZe**
>
> > **Q2**  How does RECAST balance model plasticity and stability in incremental learning? From my understanding, RECAST reparameterizes the model using lightweight coefficients and frozen template banks for each downstream task, making it a parameter-efficient fine-tuning method rather than one specifically designed for incremental learning. In the experiments, RECAST is combined with existing incremental learning methods to achieve better performance. If this understanding is correct, it would be useful to compare RECAST with other parameter-efficient fine-tuning methods, such as prompt-tuning, LoRA, and DoRA.
>
>  As discussed in an earlier response, RECAST can automatically address the stability-plasticity dilemma in incremental learning through its architectural design, not just as a parameter-efficient fine-tuning method. The balance between stability and plasticity is achieved through RECAST's core decomposition strategy: stability is maintained through shared template banks that serve as frozen knowledge repositories across tasks, preserving fundamental task-agnostic features and transformations without requiring explicit regularization or rehearsal. Plasticity is enabled through lightweight task-specific coefficients that dynamically recombine these templates that ensure adaptations for new tasks cannot interfere with existing ones.
> The table 2 in our paper compares RECAST with different LoRA variants that apply LoRA to various components of a transformer backbone. We compare against :
>  - MeLO : applies LoRA to the Q and V matrices of the Attention Block
>  - AdaptFormer: applies LoRA to the weights of the fully-connected layers of the MLP block
>  -  InfLoRA : applies LoRA to all Linear layers of a Vision Transformer
>
>  By considering your feedbacks, we have also added two more contemporary LoRA-based methods and added the results to Table 2
>  - DoRA : applies LoRA to Q, K and V matrices of the Attention Block
>  - RoSA : May be applied to both Attention and MLP layers with Sparse weight adaptation
> We also compared against a Prompt-based method called L2P which maintains a shared pool of task prompts to perform incremental learning.
> Here’s a partial snapshot of the updated table with these results :
>
> | Method                                                    | Learnt Params          | Storage     | Acc@top1(%)             |
> |-----------------------------------------------------------|------------------------|-------------|-------------------------|
> | L2P                                  | To * D * M_pool        | M + Pl      | 35.9                    |
> | InfLoRA                               | P + 2 * 10^-3 * P * (T-1) | M         | 88.5                    |
> | AdaptFormer                           | 6 * 10^-4 * Pr         | M           | 89.0                    |
> | MeLo                                  | 8 * 10^-4 * Pr         | M           | 88.7                    |
> | DoRA                                 | 1.9 * 10^-3 * Pr       | M           | 89.3                    |
> | RoSA (Full - Att.)                   | 7 * 10^-3 * Pr         | M           | 88.6                    |
> | RECAST (ours) + AdaptFormer                               | 6 * 10^-4 * Pr         | M           | 90.1 ± 0.2             |
> | RECAST (ours) + MeLo                                      | 8 * 10^-4 * Pr         | M           | 89.6 ± 0.2             |
> | RECAST (ours) + RoSA (Full - Att.)                        | 7 * 10^-3 * Pr         | M           | 88.9 ± 0.08            |
>
> As shown above, DoRA and RoSA report some benefits over LoRA variants when used alone, but are worse than those methods combined with our work while also being less parameter efficient.  That said, our results also show benefits when combined with RoSA, although also worse than LoRA variants.  This demonstrates that our benefits are more complementary than the changes to LoRA proposed by DoRA or RoSA, resulting in higher gains on our task.

---

> ### Author Response · Authors · 2024-11-23
> **Contd.(3) To Reviewer YqZe**
>
> > **W4**  It is unclear how RECAST integrates with previous incremental learning methods, as shown in Tables 1 and 2.
>
> > **Q3** How does RECAST integrate into existing incremental learning methods, as shown in Tables 1 and 2?
>
> RECAST complements existing incremental learning methods as a parameter-efficient weight generation backbone. For CNN architectures, RECAST generates base weights that PiggyBack's binary masks can modulate, and provides primary convolution weights that CLR's Ghost modules can refine. In transformer architectures, RECAST produces core weight matrices that methods like MeLO, AdaptFormer (for query/value and MLP components), DoRA (for magnitude-direction decomposition), and RoSA (for sparse adaptations) can then modify.
>
>  For example in standard LoRA, weight is reparameterized by adding a low-rank update: $W_{LoRA} = W_0 + BA$ . When integrating with RECAST, instead of using a fixed $W_0$, we use the dynamically generated weight matrix from RECAST. In Section 2.1 Equation 1, we defined that RECAST generates weights as:
> $$W_{\text{RECAST}} = \frac{1}{K}\sum_{k=1}^K\sum_{i=1}^n T_{g,i} \cdot C^k_{l,m,i}$$
> where T_(g,i) are the templates and C^k_(l,m,i) are the coefficients.
> The integrated approach combines both methods:
> $W_{\text{combined}} = W_{\text{RECAST}} + BA$
>
> Similar principles apply when integrating with other methods:
>  - CLR: $Y = \text{Ghost}(\text{Conv}(X, W_{\text{RECAST}}))$
>  - PiggyBack: $W_{\text{combined}} = W_{\text{RECAST}} \odot M$ where M is a binary mask matrix.
>  - RoSA: $ W_{\text{combined}} = W_{\text{RECAST}} + S \odot W_{\text{RECAST}} +  (BA)$ where S is a sparse binary mask applied to the LoRA update.
>
> These integrations maintain RECAST's own reparameterization and task adaptations benefits while leveraging the unique benefits of each method. For empirical validation of these combinations, see Tables 1 and 2 in the main paper.
> We briefly noted this integration in the `Implementation Details` paragraph of Section 3. Additionally, this expanded discussion is now also included in the Appendix Section 4.2

---

> > ### Author Response · Authors · 2024-11-30
> > **Official Follow-up comment by Authors**
> >
> > We would like to express our appreciation to the reviewer for identifying important technical and analytical aspects of our study. Your reviews helped us enhance the presentation and critical analysis of our work.
> >
> > Given the remaining time, we are happy to provide any further clarification on any of your additional queries that might help improve the score.

---

> > > ### Comment · Reviewer_YqZe · 2024-12-03
> > >
> > > Thank the authors for providing a thorough response to address my concern. I have raised my score accordingly.

---

### Official Review · Reviewer_GjwD · 2024-11-04

**Soundness:** 3
**Presentation:** 2
**Contribution:** 3
**Rating:** 6
**Confidence:** 3

**Summary:**

This paper addresses the problem of continual learning with the goal of creating a learning pipeline that reduces the number of learnable parameters, making it feasible for resource-limited devices. To achieve this, the authors propose a Neural Mimicry pipeline, which decomposes model weights into a bank of templates. Using these frozen templates, the model only learns coefficients for subsequent downstream tasks. Experimental results highlight the potential of the proposed method. However, I have several questions, and my detailed comments are as follows.

**Strengths:**

The proposed idea of incorporating Neural Mimicry into continual learning is novel and interesting. By decomposing model weights into frozen template banks and focusing solely on learning coefficients, the approach significantly reduces the number of learnable parameters. This paper introduces a new pipeline and valuable insights for the continual learning community.

Experiments on both CNN and ViT-small models demonstrate the effectiveness of the proposed method and highlight its architecture-agnostic nature.

**Weaknesses:**

The writing in this paper could be improved. For the pseudo-code, it would enhance readability to introduce the notations alongside the code for easier understanding. Additionally, more details on how to obtain the pre-trained model weights should be clearly provided.

The effectiveness of the method on larger models and datasets has not yet been adequately addressed.

**Questions:**

How are the pre-trained weights obtained for further Neural Mimicry decomposition? This seems quite important for the algorithm’s performance. If the model is pre-trained on data that differs significantly from the sequential tasks that follow, learning only coefficients may be insufficient, in my view.

In Table 1, why the total storage for RECAST + Piggyback/CLR increase linearly? For CNN-based models, why can’t this be further reduced, given that the learnable task parameters are already minimized from  $P$  to  $4 \times 10^{-3} P$ ?

How does the algorithm perform with larger models and datasets, such as ViT-base/Large with the ImageNet dataset? Would the Neural Mimicry method still be effective? Further discussion on this would be beneficial.

In Table 2, why does the total storage remain unchanged? As more tasks are added, shouldn’t total storage also increase with  $M + T \times 6/8 \times 10^{-4} Pr$ ?

Efficiency in training/learning on edge devices is also important. It would be highly valuable to include a comparison of wall-clock training time and GPU memory in Tables 1 and 2. Given the significantly reduced number of learnable parameters, the proposed method should enhance training efficiency, which could be seen as an advantage of this approach, but empirical evidence is needed to confirm this.

It would also be helpful to report the training complexity of the Neural Mimicry process.

Finally, in Table 1, why do the accuracies for EWC, LWF, and GDUMB perform so much worse than ResNet-34?

---

> ### Author Response · Authors · 2024-11-23
> **To Reviewer GjwD**
>
> We deeply appreciate the reviewers effort and providing us the opportunity to elaborate on several important aspects of our work. Below we provide our responses by breaking it down by the points presented by the reviewer.
>
> > **W1** The writing in this paper could be improved. For the pseudo-code, it would enhance readability to introduce the notations alongside the code for easier understanding. Additionally, more details on how to obtain the pre-trained model weights should be clearly provided.
>
>   We welcome the opportunity to enhance the readability by making the following changes to incorporate your suggestions
>  - Introducing the RECAST overview earlier with Figure 2 for better reference during reading
>  - Modified the algorithms with notation definitions
>  - We have rearranged Section 2 to provide a better flow of concepts and parseability of the different components. We have also consolidated related works into the introductory part of Section 2,  to contextualize each component of RECAST before going into details. We have likewise, removed the redundant references from the methodology subsections.
>  - We now explicitly mention how the pretrained models (trained on ImageNet21k dataset) have been obtained from the official Python timm library  in Appendix section A.3
>
> > **Q1** How are the pre-trained weights obtained for further Neural Mimicry decomposition? This seems quite important for the algorithm’s performance. If the model is pre-trained on data that differs significantly from the sequential tasks that follow, learning only coefficients may be insufficient, in my view
>
>  - We obtained the ViT and DiNO weights from the official timm repository [1]. More implementation details are covered in Section 3, as well as direct links are provided in Appendix section A.3. ResNet weights were obtained from Pytorch Hub[2]. All backbones are standard and easily accessible. The Resnet weights are pretrained on the ImageNet1k dataset. The ViT weights are generated by pretraining on ImageNet1k, and then finetuning them on the Imagenet21k dataset. The DiNO ViT pretraining follows the specifics from their original paper, without a classifier head[3].
>  - In Table 6 we show that the performance of the reconstructed models match their reported Imagenet1k accuracy@top1 scores in their respective repositories. Which means, all our baselines and RECAST models start from the same pretrained backbone. Additionally, as we discuss further later, in Fig. 4 we show that we can further improve the backbones through coefficient tuning.  These results demonstrate that our approach generalizes across multiple architectures and pretraining datasets.
> ```
> [1] Ross Wightman. Pytorch image models. https://github.com/rwightman/
> pytorch-image-models, 2019
> [2] “Resnet34 — Torchvision Main Documentation.” Pytorch.org, pytorch.org/vision/main/models/generated/torchvision.models.resnet34.html.
> [3] Caron, Mathilde et al. “Emerging Properties in Self-Supervised Vision Transformers.” 2021 IEEE/CVF International Conference on Computer Vision (ICCV) (2021): 9630-9640.
> ```
> > **Q2** In Table 1, why the total storage for RECAST + Piggyback/CLR increase linearly? For CNN-based models, why can’t this be further reduced, given that the learnable task parameters are already minimized from P to 4×10−3P ?
>
>  Both CLR and Piggyback are adapter-based methods. CLR stores taskwise Ghostmodules applied after each convolution layer, and Piggyback stores taskwise binary masks applied over the entire weight. In LoRA-like approaches, the low-rank update matrices can be mathematically merged with the existing backbone weights - keeping the total storage unchanged. This is not possible for CLR GhostModules, and they need to be stored as extra adapters for each task. However, this is also possible for the Binary masking approaches like PiggyBack. We have updated the notations in Table 1 to reflect this and further improve clarity. We added a partial snapshot of the results below for your reference.  A brief note is also made in Section 3’s Comparison of Resource Efficiency paragraph to further clarify the reported metrics.
>
> | Method                          | Learnt Params    | Storage     | Acc@top1% |
> |---------------------------------|------------------|-------------|-----------|
> | Piggyback                       | P                | M           | 82.9      |
> | CLR                             | 4 * 10^-3 P      | M + G       | 79.9      |
> | RECAST (ours) + Piggyback       | P                | M           | 83.9      |
> | RECAST (ours) + CLR             | 4 * 10^-3 P      | M + G       | 80.9      |

---

> ### Author Response · Authors · 2024-11-23
> **Contd. To Reviewer GjwD**
>
> > **W2** and **Q3** The effectiveness of the method on larger models and datasets has not yet been adequately addressed.
> How does the algorithm perform with larger models and datasets, such as ViT-base/Large with the ImageNet dataset? Would the Neural Mimicry method still be effective? Further discussion on this would be beneficial
>
>  Thank you for raising this important point about scalability. We did actually compare to ViT-Small (21M parameters), ViT-Base (86M parameters), ViT-Large (307M parameters), and the small DINO-ViT in the appendix, in addition to a ResNet-34.  Due to your comment, we have moved these results to Fig. 4 of our updated paper, where we show that our RECAST approach boosts performance across each architecture and pretrained weights on the six datased used in our incremental learning task.  We also have an expanded discussion on our Neural Mimicry approach over these models in Tab. 6 in the Appendix Section A.3 and reproduced in our next response.  These results coupled with our results using a ResNet-34 well demonstrate RECAST’s ability to generalize.  That said, we would also like to highlight that on edge devices where our compact parameter representation presents the most attractive application, large models are not computationally feasible. We appreciate the opportunity to clarify this aspect of our work.
>
>
> > **Q4** In Table 2, why does the total storage remain unchanged? As more tasks are added, shouldn’t total storage also increase with M+T×6/8×10−4Pr?
>
> The LoRA-based methods in Table 2 leverage a key architectural advantage - after training, their task-specific parameters can be mathematically merged into the base model weights through low-rank matrix updates. This means while we train T×6/8×10^-4P parameters per task, the final storage remains M (backbone size). In contrast, adapter methods in Table 1 require explicit storage.  For example, the CLR method learns a GhostModule after each CNN layer, which is done for each task.
>
> We appreciate you helping us elaborate on this distinction in the table presentation. By taking your suggestions into account we have modified both of the tables to explicitly mention that the number of parameters and storage are reported per task. We also added a note in the paper to explain that this is due to merging the low-rank updates. We also refined the table notations for further clarity.

---

> ### Author Response · Authors · 2024-11-23
> **Contd.(2) To Reviewer GjwD**
>
> > **Q5** Efficiency in training/learning on edge devices is also important. It would be highly valuable to include a comparison of wall-clock training time and GPU memory in Tables 1 and 2. Given the significantly reduced number of learnable parameters, the proposed method should enhance training efficiency, which could be seen as an advantage of this approach, but empirical evidence is needed to confirm this.
>
>  In the following table, we report the throughput and latency of two representative baseline models and their RECAST-adapted models during inference. For inference, the RECAST weights can be precomputed and cached. The evaluation has been performed on a single RTX 6000 GPU, and the metrics were averaged across [2, 26, 32, 128, 256] batch sizes. From the table, it is clear that all the RECAST models in general demonstrated higher throughput and latency compared to their baseline counterparts. These metrics directly reflect the model's performance during deployment, which is a critical factor in real-world applications.
>
> | Model       | Throughput (samples/sec) | Latency P95 (ms) |
> |-------------|--------------------------|------------------|
> | RECAST      | 2707.2                   | 1.092            |
> | LoRA        | 1956.2                   | 1.153            |
> | RECAST+LoRA | 2008.5 (+2.7\%)          | 1.114 (-3.4\%)   |
> | RoSA        | 1166.9                   | 1.644            |
> | RECAST+RoSA | 1217.5 (+4.3\%)          | 1.613 (-1.9\%)   |
>
> We want to note that, the best benefits of RECAST-adapted models would be in federated learning on edge devices. Earlier studies demonstrate that the low parameter footprint of RECAST is a key advantage in these environments[1], even if the direct performance impact is not easily quantifiable in a centralized setting[ 2]. Given the limited time of the rebuttal, we haven’t reported the metrics here - but we hope to include the full table in camera-ready version.
> ```
> [1] Ernie Chan, Marcel Heimlich, Avi Purkayastha, and Robert Van De Geijn. Collective communication: theory, practice, and experience. Concurrency and Computation: Practice and Experience, 19(13): 1749–1783, 2007.
> [2]  Bryan A. Plummer, Nikoli Dryden, Julius Frost, Torsten Hoefler, and Kate Saenko. Neural parameter allocation search, 2022
> ```
>
> > **Q6**  It would also be helpful to report the training complexity of the Neural Mimicry process.
>
>  We appreciate the suggestion to include these metrics as they highlight an important practical aspect of our method. We're pleased to share a detailed analysis of the reconstruction process across model scales. This is also now included in the Appendix Section A.3 Table 5 (and provided below)
> | Model (size)      | Peak GPU | CPU Usage | Per Epoch | Total Time |
> | ----------------- | -------- | --------- | --------- | ---------- |
> | ViT-Small (~21M)  | 0.33GB   | 3.75%     | 36.2ms    | 37.40s     |
> | ViT-Base (~86M)   | 1.2GB    | 3.70%     | 112ms     | 102.3s     |
> | ViT-Large (~307M) | 4.5GB    | 3.70%     | 340ms     | 245.0s     |
>
> The results demonstrate that Neural Mimicry has a modest memory footprint (0.33-4.5GB GPU memory) with minimal CPU overhead (~3.7%). It also linearly scales with model size. Furthermore, the complete reconstruction completes in minutes even for large models. Notably, this step achieves 98-100% reconstruction accuracy (Table 6) and requires significantly less resources than model pretraining.  A snapshot of Table 6 is provided below:
>
> | Model                             | Imagenet   | Reconstruction |
> |-----------------------------------|------------|----------------|
> |                                   | Acc@top1   | Similarity     |
> | Resnet-34                         | 73.3       | -              |
> | Recon. Resnet-34                  | 73.0       | 99\%           |
> | ViT-Small                         | 74.6       | -              |
> | Recon. ViT-Small (MLP)            | 74.6       | 100\%          |
> | Recon. ViT-Small (VAE + MSE)      | 73.8       | 98.5\%         |
> | Recon. ViT-Small (VAE + SmoothL1) | 73.0       | 98.6\%         |
> | ViT-Base                          | 81.1       | -              |
> | Recon. ViT-Base                   | 81.0       | 100\%          |
> | ViT-Large                         | 84.4       | -              |
> | Recon. ViT-Large                  | 84.4       | 100\%          |
> | DINO Small                        | 77.0       | -              |
> | Recon. DiNO Small                 | 76.9       | 99\%           |

---

> ### Author Response · Authors · 2024-11-23
> **Contd.(3) To Reviewer GjwD**
>
> > **Q7** Finally, in Table 1, why do the accuracies for EWC, LWF, and GDUMB perform so much worse than ResNet-34?
>
>  The traditional Incremental Learning methods like EWC, LWF and GDUMB suffer from catastrophic forgetting due to task interference. While they manage to maintain performance in TIL on homogenous datasets like MNIST, these methods perform very poorly when the tasks are from diverse domains with minimal semantic overlap. This observation is further corroborated by existing studies[1, 2, 3, 4], highlighting the limitations of these methods. In contrast, classifier-only fine-tuning and RECAST perform better primarily because they maintain stable feature representations through a frozen backbone while not explicitly attempting to preserve previous task performance, which proves to be a more effective strategy for handling such diverse task sequences.
> ```
> [1] Rui Yang, Matthieu Grard, Emmanuel Dellandréa, Liming Chen. Entropy-Guided Self-Regulated Learning Without Forgetting for Distribution-Shift Continual Learning with blurred task boundaries. 2023. ￿hal-04228875￿
> [2] Ha D, Kim M, Jeong CY. Online Continual Learning in Acoustic Scene Classification: An Empirical Study. Sensors (Basel). 2023 Aug 3;23(15):6893. doi: 10.3390/s23156893. PMID: 37571676; PMCID: PMC10422258.
> [3] Yunhao Ge, Yuecheng Li, Shuo Ni, Jiaping Zhao, Ming-Hsuan Yang, and Laurent Itti. Clr: Channelwise lightweight reprogramming for continual learning, 2023
> [4] Guo, Yiduo et al. “Dealing with Cross-Task Class Discrimination in Online Continual Learning.” 2023 IEEE/CVF Conference on Computer Vision and Pattern Recognition (CVPR) (2023): 11878-11887
> ```

---

> > ### Author Response · Authors · 2024-11-30
> > **Official Follow-up comment by Authors**
> >
> > We would like to express our appreciation to the reviewer for identifying important technical and analytical aspects of our study. Your reviews helped us enhance the presentation and critical analysis of our work.
> >
> > Given the remaining time, we are happy to provide any further clarification on any of your additional queries that might help improve the score.

---

> > ### Comment · Reviewer_GjwD · 2024-12-03
> >
> > Thanks for the authors' response. Most of my concerns have been addressed and I keep my original rating.

---

### Official Review · Reviewer_KSLE · 2024-11-05

**Soundness:** 2
**Presentation:** 2
**Contribution:** 2
**Rating:** 5
**Confidence:** 4

**Summary:**

This paper proposes a Neural Mimicry method to study a very small set of parameters for continual learning purposes. It studies both ViT and ResNet applications. It showed performance benefits over several benchmarks.

**Strengths:**

- The paper studies an important task.
- Motivating towards neural mimic and learnings from proxy domains are helpful in solving the CL tasks.

**Weaknesses:**

- Experiments are very small scale. Two target models are only ~21M in size (Ln 305), and does not demonstrate scalability to large models.
- No forgetting values are presented that are critical aspects of CL learning setups.
- Fig. 1 is misleading and hard to get the 2*10^-6 parameters - instead this is a ratio instead of parameters. Moreover, it's not thoroughly shown this is a parameter that scales across networks. Would it be the same for LLM? Otherwise stating the method yields <<<1% can be fairly misleading, and in fact wrong if the units are parameters.

**Questions:**

As above.

---

> ### Author Response · Authors · 2024-11-23
> **To Reviewer KSLE**
>
> We thank and appreciate the reviewer for their valuable feedbacks on our work. Below we provide our responses by breaking it down by the points presented by the reviewer.
>
> > **W1** Experiments are very small scale. Two target models are only ~21M in size (Ln 305), and does not demonstrate scalability to large models.
>
> Thank you for raising this important point about scalability. We did actually compare to ViT-Small (21M parameters), ViT-Base (86M parameters), ViT-Large (307 parameters), and the small DINO-ViT in the appendix.  Following your comment, we have moved these results to Fig. 4 of our updated paper and added an expanded discussion on our Neural Mimicry approach over these models in Tab. 6 in the Appendix.  Both the table and the figure show that our RECAST approach boosts performance on the six datasets of our experiments across each architecture and pretrained weights and coupled with experiments using a ResNet-34, well demonstrate RECAST's ability to generalize.  That said, we would also like to highlight that on edge devices where our compact parameter representation presents the most attractive application, large models are not computationally feasible. We appreciate the opportunity to clarify this aspect of our work.
>
> > **W2** No forgetting values are presented that are critical aspects of CL learning setups.
>
> Thank you for the opportunity to clarify this important benefit of our approach.  Specifically, RECAST does not forget any prior knowledge as all task specific parameters like the mixing coefficients C from Eq. 1 are learned for each task in isolation, whereas the templates T are kept frozen.  Thus, once C are learned we can always reconstruct their exact weights even when we train new coefficients for a new task. This behavior is analogous to reprogramming methods like CLR[1],  Piggyback[2], SUPSUP[3], and LoRA variants[4, 5] , which makes traditional forgetting metrics that measure interference between tasks not meaningful as all task-specific parameters are independently maintained, completely avoiding any catastrophic forgetting.
> ```
> [1] Yunhao Ge, Yuecheng Li, Shuo Ni, Jiaping Zhao, Ming-Hsuan Yang, and Laurent Itti. Clr: Channelwise lightweight reprogramming for continual learning, 2023
> [2] Michael McCloskey and Neal J. Cohen. Catastrophic interference in connectionist networks: The sequential learning problem. volume 24 of Psychology of Learning and Motivation, pp. 109–165.
> [3] Mitchell Wortsman, Vivek Ramanujan, Rosanne Liu, Aniruddha Kembhavi, Mohammad Rastegari, Jason Yosinski, and Ali Farhadi. Supermasks in superposition. Advances in Neural Information Processing Systems, pages 1517315184, 2020.
> [4] Shih-Yang Liu, Chien-Yi Wang, Hongxu Yin, Pavlo Molchanov, Yu-Chiang Frank Wang, Kwang-Ting Cheng, and Min-Hung Chen. DoRA: Weight-decomposed low-rank adaptation. In Ruslan Salakhutdinov, Zico Kolter, Katherine Heller, Adrian Weller, Nuria Oliver, Jonathan Scarlett, and Felix Berkenkamp (eds.), Proceedings of the 41st International Conference on Machine Learning, volume 235 of Proceedings of Machine Learning Research, pp. 32100–32121. PMLR, 21–27 Jul 2024b.
> [5]Mahdi Nikdan, Soroush Tabesh, Elvir Crnˇcevi´c, and Dan Alistarh. RoSA: Accurate parameter-efficient fine-tuning via robust adaptation. In Ruslan Salakhutdinov, Zico Kolter, Katherine Heller, Adrian Weller, Nuria Oliver, Jonathan Scarlett, and Felix Berkenkamp (eds.), Proceedings of the 41st International Conference on Machine Learning, volume 235 of Proceedings of Machine Learning Research, pp. 38187–38206. PMLR, 21–27 Jul 2024.
> ```

---

> ### Author Response · Authors · 2024-11-23
> **Contd. To Reviewer KSLE**
>
> > **W3** Fig. 1 is misleading and hard to get the 2*10^-6 parameters - instead this is a ratio instead of parameters. Moreover, it's not thoroughly shown this is a parameter that scales across networks. Would it be the same for LLM? Otherwise stating the method yields <<<1% can be fairly misleading, and in fact wrong if the units are parameters
>
>  We appreciate the suggestion to make Figure 1's labeling more precise, and we have updated the figure to reflect that these are parameter ratios rather than absolute counts.
>  - The absolute values are more explicitly reported in figure 3 . We also want to clarify that our parameter claims are precise and well-supported in Tables 1 and 2 explicitly show our parameter counts (i.e: "3 × 10^-6 P" for ResNet34 and "2 × 10^-6 P" for 12-layer ViT Where P is the total parameters with a single set of coefficients).
>  - Besides this, we have further elaborated on how coefficients can be parametrically scaled in subsection 2.3 as a combination of total layers, number of target modules, coefficient set, and sharing scheme. For example, consider a LLaMa-3 model with 32 layers and 8 billion parameters, where we apply RECAST to the MLP block which has 3 fully connected projection layers (UP, DOWN, and GATE). Thus, if we apply a sharing scheme of 16 groups with 2 templates per bank and 1 set of coefficients per projection module we will have exactly 192 parameters which translates to approximately 2 * 10^-6 % parameters. Thus, after Decomposition and Neural Mimicry, we will have to finetune <<<1% coefficients per task.  As LLMs are also built using the same building blocks that we have discussed in our study, we can surmise that RECAST can also be applied to LLMs.

---

> ### Author Response · Authors · 2024-11-30
> **Official Follow-up comment by Authors**
>
> We would like to express our appreciation to the reviewer for identifying important technical and analytical aspects of our study. Your reviews helped us enhance the presentation and critical analysis of our work.
>
> Given the remaining time, we are happy to provide any further clarification on any of your additional queries that might help improve the score.

---

> > ### Comment · Reviewer_KSLE · 2024-12-02
> > **Reviewer Response**
> >
> > After reading the author response I am still not fully convinced that the paper is ready. I acknowledge the clarifications made by the authors, and would raise score and finalize recommendation at slightly below the bar.

---

> > > ### Author Response · Authors · 2024-12-03
> > >
> > > Thank you Reviewer KSLE for updating your review! While we are happy you raised your score we are sad to see you did not recommend acceptance. We would appreciate it if you gave the justification for your recommendation as it’ll be helpful to know where we can improve as well as for the area chair to make their recommendations. Thanks again!

---

### Author Response · Authors · 2024-11-24
**Official Comment by Authors to the reviewers**

We thank the reviewers for their thorough and constructive feedback. They identified several key technical strengths of our work:

- Parameter Efficiency: RECAST achieves extreme parameter efficiency (3×10^-6 of model size for ResNet34, 2×10^-6 for ViT) while maintaining performance, addressing a critical need for resource-constrained environments (highlighted by KSLE and GjwD)

- Strong Empirical Results: Our approach improves accuracy by up to 3% over state-of-the-art methods across architectures, with particularly strong gains when combined with existing methods like AdaptFormer (90.1%), MeLO (89.6%), and RoSA (88.9%) (highlighted by YqZe and uU62 )

- Architectural Innovation: The Neural Mimicry reconstruction pipeline enables fast (< 5 minutes) and accurate (98-100% reconstruction similarity) adaptation of pretrained models without expensive retraining, addressing important efficiency concerns raised by GjwD regarding training complexity and resource usage.

- Practical Applicability: RECAST demonstrates consistent improvements across 6 diverse datasets and multiple architectures (ResNet, ViT-Small/Base/Large, DINO), while seamlessly integrating with and enhancing existing parameter-efficient methods (highlighted byuU62)


Primary reviewer concerns and our responses include :

1. Scalability and Large Model Experiments (KSLE, GjwD):
We provided extensive additional results demonstrating RECAST's effectiveness across model scales (ViT-Small: 21M, Base: 86M, Large: 307 M parameters) with consistent 98-100% reconstruction accuracy and improved downstream performance. These results are now prominently featured in Fig. 4.

2. Resource Efficiency & Implementation Details (YqZe, uU62):
We added detailed analysis of computational requirements (Table 5), showing Neural Mimicry completes in minutes with minimal overhead (0.33-4.5GB GPU, ~3.7% CPU). We also clarified RECAST's integration with existing methods through new equations and implementation details in Section A.4.2.

3. Reorganization and Clarity (YqZe, GjwD):
We restructured Section 2 to improve flow and readability, moved the RECAST overview (Fig. 2) earlier, and added expanded discussions on technical contributions and comparisons to prior work.

4. Performance Analysis (uU62, GjwD):
We provided comprehensive task-wise performance breakdown (Table 4), demonstrating consistent improvements across all datasets and showing RECAST's robustness to task ordering effects.

We are happy to accommodate further reviews to enhance the quality of the work.

---

### Meta-Review · Area_Chair_9sVo · 2024-12-21

**Metareview:**

The paper proposes an incremental learning framework, where only co-efficients of a basis function is learnt for new tasks. These basis functions are called a template bank and it consists of fixed and pre-trained templates shared across tasks, In order to convert a given network to this format, the paper proposes a method called  Neural Mimicry that reparameterizes model weights by decomposing them into a shared template bank and lightweight task-specific coefficients. Experimental results validate the method’s potential for enabling continual learning. All the reviewers like parameter efficiency of the proposed method as well as the novelty of the approach. The weakness of the methods was attributed to presentation quality as well as the lack of large scale experiments. The authors provided extensive rebuttal that most of the reviewers were happy with.

**Additional Comments On Reviewer Discussion:**

All reviewers provided responses to the author rebuttals which was long and thorough. The quality of the rebuttal  improved the ratings across most reviewers except reviewer KSLE.

---

### Decision · Program_Chairs · 2025-01-22

Accept (Poster)